METHODS AND RESOURCES

# High-throughput characterization of *Mycobacterium tuberculosis* gene function across diverse conditions

Kayla M. Dinshaw[1], Katie A. Lien[1], Matthew Knight[2], Sorel V. Yimga Ouonkap[1], Hualan Liu[3], David F. Savage[1,4], Hans K. Carlson[5], Adam M. Deutschbauer[2,5], Sarah A. Stanley [1]*

1 Department of Molecular and Cell Biology, University of California, Berkeley, California, United States of America, 2 Department of Plant and Microbial Biology, University of California, Berkeley, California, United States of America, 3 Joint Genome Institute, Lawrence Berkeley National Laboratory, Berkeley, California, United States of America, 4 Howard Hughes Medical Institute, University of California, Berkeley, California, United States of America, 5 Environmental Genomics and Systems Biology Division, Lawrence Berkeley National Laboratory, Berkeley, California, United States of America

* sastanley@berkeley.edu

## Abstract

*Mycobacterium tuberculosis* (Mtb) is a human bacterial pathogen that establishes chronic infection in the lung. Although the genome of Mtb was sequenced nearly 25 years ago, the genetic basis of Mtb's success as a human pathogen remains to be fully elucidated. Large-scale mutation-based genetic approaches to understanding gene function are hindered by the limited throughput of traditional transposon sequencing strategies used in mycobacteria. To create a resource for determining the function of genes, we generated a pooled random barcode transposon-site sequencing (RB-TnSeq) library in Mtb. A unique 20-nucleotide barcode in the transposon allows for rapid, high-throughput genetic screening without the laborious protocol of standard bacterial TnSeq screens. We performed 95 RB-TnSeq screens on an array of carbon sources, nitrogen sources, stressors, and antibiotics. Using the resulting dataset, we examined phenotypes of *pe* and *ppe* genes, a mycobacterial gene family whose function has long been elusive, uncovering 187 novel phenotypes across 37 genes in this family. We propose a pathway for lactate utilization in which the ESX-5 type VII secretion system may export PPE3, facilitating the import of D- and L-lactate into the bacterial cell. Notably, we identify a candidate D-lactate dehydrogenase that may mediate this metabolic capability. Additionally, we find that the proton-pumping NADH dehydrogenase Nuo is required for utilization of propionate, highlighting the metabolic flexibility of Mtb. Lastly, we characterize a novel mutant that confers resistance to the new tuberculosis antibiotic pretomanid. Results from these genetic screens will facilitate the development of additional new hypotheses about the function of uncharacterized genes and will expand our knowledge of Mtb metabolism and resistance to stress.

**Data availability statement:** All data will be contained within the supplementary files of the paper and can be browsed on a publicly available website: https://fit.genomics.lbl.gov/cgi-bin/org.cgi?orgId=MycoTube. Data can also be found on FigShare:https://doi.org/10.6084/m9.figshare.30444569.

**Funding:** This work was supported by NIH 1R01AI143722 to SAS (https://www.niaid.nih.gov), RM1 GM135102 to AD (https://www.nigms.nih.gov), and HHMI to DFS (https://www.hhmi.org). The funders had no role in study design, data collection and analysis, decision to publish, or preparation of the manuscript.

**Competing interests:** I have read the journal's policy and the authors of this manuscript have the following competing interests. SAS. is on the scientific advisory board of Xbiotix Therapeutics, an antimicrobials company whose work has no overlap with this study. DFS. is a founder and on the scientific advisory board of Scribe Therapeutics, a CRISPR company whose work has no overlap with this study.

**Abbreviations:** CFU, colony-forming units; DMEM, Dulbecco's Modified Eagle Medium; FBS, fetal bovine serum; MDR, multidrug-resistant; Mtb, *Mycobacterium tuberculosis*; PDIM, phthiocerol dimycocerosate; PGRS, polymorphic GC-rich sequences; RB-TnSeq, random barcode transposon-site sequencing; TLC, thin-layer chromatography; TnSeq, transposon insertion sequencing; WT, wild-type; XDR, extensively drug-resistant.

## Introduction

*Mycobacterium tuberculosis* (Mtb) is a bacterium that infects the lungs and is the causative agent of tuberculosis, which is estimated to infect a quarter of the global population [1]. The success of Mtb as a human pathogen may be attributed to many unique features of mycobacterial physiology. Mtb resides in the phagosome of macrophages, which activate an arsenal of host defenses to attempt to eradicate the infection. For instance, the phagosome is acidified by the vacuolar-H(+)-ATPase, NOS2 catalyzes production of nitric oxide, and the NADPH oxidase generates a respiratory burst of superoxide. Although these immune processes are successful at killing many pathogens, Mtb can persist and establish a chronic infection.

To grow inside the host, Mtb must assimilate host-derived macromolecules including fatty acids, amino acids, and carbohydrates to use as carbon and nitrogen sources. A large portion of the Mtb genome is seemingly dedicated to lipid metabolism [2]. Consistent with this, mutants lacking the fatty acid transporter *mce1* or the cholesterol importer *mce4* exhibit attenuated virulence in murine models of infection [3,4], supporting the idea that fatty acids and cholesterol are important in vivo carbon sources. However, the range of carbon sources used by Mtb during infection remains unresolved. For instance, Mtb grows in vitro on L-lactate [5,6], a metabolite produced from the metabolism of carbohydrates. L-lactate is present in guinea pig granulomas [7] and is produced by immune cells during Mtb infection due to the Warburg effect, the metabolic shift from oxidative phosphorylation to aerobic glycolysis [8–11]. An L-lactate dehydrogenase mutant in Mtb is attenuated for infection in human macrophages [5] and is under positive selection in Mtb clinical strains [6], suggesting that molecules other than lipids may act as carbon sources in vivo. Interestingly, there is little research into whether Mtb can use the stereoisomer D-lactate as a carbon source.

This metabolic flexibility of Mtb reflects its ability to survive hostile conditions, which also extends to antibiotic exposure. Mtb is both intrinsically resistant to many antibiotics and acquires de novo resistance to clinically used antibiotics. As a result, antibiotic therapy for TB requires combinatorial therapy that presents significant side effects. Depending on the bacterial strain, treatment can range from three to nine months in duration [12], contributing to treatment disruption and the rise of multidrug-resistant (MDR) and extensively drug-resistant (XDR) strains. Drug-susceptible TB regimens include antibiotics such as isoniazid, rifampicin, pyrazinamide, and ethambutol. In response to the rise of MDR and XDR Mtb strains, WHO updated their guidelines in 2022 to include the BPaLM regimen [13], consisting of bedaquiline, pretomanid, linezolid, and moxifloxacin. Pretomanid, also referred to as PA-824, is a nitroimidazole that inhibits the synthesis of cell envelope ketomycolates [14]. The full spectrum of mutations that confer antibiotic resistance or susceptibility is yet to be determined for most antibiotics, particularly for newer drugs like pretomanid. Thus, the ability to predict what mutations may arise could be a clinically powerful tool.

Mtb encodes ~4,000 genes, a similar size to the model bacterium *Escherichia coli* [15]. However, many genes remain uncharacterized and are annotated as encoding "hypothetical proteins" or annotated based on homology [16]. The requirement for

high-containment facilities and the slow growth of Mtb (21 days to grow a colony on solid agar) complicate laboratory experiments. In bacteria, genetic screens are often accomplished using transposon insertion sequencing (TnSeq), which couples pooled transposon insertion mutagenesis with next-generation sequencing [17–20]. Transposon-mutant libraries are exposed to a condition of interest, and the enrichment or depletion of transposon mutants are used to characterize gene function. TnSeq screens in Mtb have revealed essential genes in broth [21–23] and genes important for virulence in various mouse models [24–26]. TnSeq screens have also been conducted in biologically-relevant conditions, some of which include acid and oxidative stress [27], tuberculosis antibiotics [28], hypoxia [29], and growth on cholesterol [22]. A major limitation of TnSeq methods are the long and laborious protocols for preparing the sequencing libraries.

To address the sample preparation limitations of TnSeq in bacteria, random barcode transposon-site sequencing (RB-TnSeq) was developed and has since been applied to diverse environmental and commensal bacterial species [30–32]. In RB-TnSeq, each transposon contains a unique 20 nucleotide barcode. One round of traditional TnSeq is performed on the initial library to identify the genetic location of each transposon insertion and its associated barcode. Then, barcode abundance can simply be quantified by PCR and deep sequencing of DNA barcodes (BarSeq), allowing for rapid, high-throughput screening.

In this study, we generated an RB-TnSeq library in Mtb and systematically profiled genetic requirements across a diverse panel of carbon sources, nitrogen sources, immune-relevant stressors, and antibiotics. The resulting dataset enabled functional insights into the enigmatic *pe* and *ppe* family genes, which encode a mycobacterial protein family named after their conserved proline and glutamate motifs. These findings reveal a lactate utilization pathway in which ESX-5 may secrete PPE3, facilitating the uptake of lactate, where it is converted into pyruvate by stereospecific lactate dehydrogenases. Additionally, we find that the NADH dehydrogenase Nuo is essential for propionate utilization, offering a metabolic insight into why Mtb encodes three distinct NADH dehydrogenases. We also identify a putative operon involved in resistance to pretomanid, a recently approved antibiotic for treatment of drug resistant Mtb infections. These high-throughput genetic screens uncover novel aspects of Mtb biology. Elucidating molecular mechanisms behind drug susceptibility and resistance, metabolic flexibility, and resistance against immune stressors will be imperative for developing new therapeutics to combat the global tuberculosis epidemic.

## Results

### High-throughput fitness data with the Mtb RB-TnSeq library

To create an RB-TnSeq library in Mtb, we ligated the temperature-sensitive mycobacteriophage vector phAE159 with pMtb_NN1, a plasmid containing a transposase under the T6 mycobacterial promoter, a *Himar1* mariner transposon with unique 20-nucleotide barcodes, and a kanamycin resistance cassette (S1 Fig). The resulting phagemid was electroporated into *Mycobacterium smegmatis* and incubated at 30 °C for lytic phage production. The wild-type (WT) Mtb strain H37Rv was then transduced with the mycobacteriophage at 37 °C, the non-lytic temperature. The resulting Mtb colonies were pooled, and TnSeq was performed to map the barcoded transposons to their location within the genome. This analysis revealed approximately 600 essential genes (S1 Table), consistent with previous findings [21,33]. However, because identical barcodes can be associated with multiple transposon insertion sites, not all barcodes are usable. Consequently, the RB-TnSeq library contains ~60,000 unique barcoded transposon mutants across 2,881 genes. As TnSeq libraries in Mtb typically represent ~3,300 genes [21,22], this RB-TnSeq library is not fully saturated with usable barcodes. Construction of the Mtb RB-TnSeq library was technically challenging, largely due to the loss of barcode diversity at each step in the lengthy protocol. Following optimization of the phagemid electroporation (see Materials and methods) and scale-up of phage production, we proceeded with using the ~60,000 barcode library to conduct genetic screens.

To capitalize on the high-throughput nature of RB-TnSeq, we sought to conduct genetic screens across a chemical library of carbon sources, nitrogen sources, antibiotics, and stressor compounds. First, we tested if chemical library compounds met the growth parameters to conduct an RB-TnSeq screen. Carbon and nitrogen sources were dissolved in

a modified Sauton's minimal media, such that each carbon or nitrogen source was the predominant nutrient. A carbon or nitrogen source compound was used for RB-TnSeq screening if a minimal threshold of ~3 doublings on the nutrient was observed (S2 and S3 Tables). For antibiotics and stressors, IC50s were measured and an RB-TnSeq screen was then performed if the compound was inhibitory towards Mtb in the range of concentrations measured (S4 Table). In total, we tested 100 carbon sources, including sugars, nucleotides, amino acids, and lipids, 43 nitrogen sources, including amino acids and nucleotides, and 131 stressors and antibiotics.

After RB-TnSeq cultures reached the growth parameters on their given compound, cultures were pelleted for genomic DNA extraction, followed by PCR amplification, and deep sequencing of DNA barcodes (BarSeq). The number of barcodes at the end of the experiment was compared to the number of barcodes at the beginning of the experiment, which is represented by the log2 fold change or "BarSeq fitness." Barcoded transposon mutants that confer a growth advantage have a positive BarSeq fitness, while barcoded transposon mutants that confer a growth disadvantage have a negative BarSeq fitness (Fig 1A). A t-like statistic was computed that accounts for the consistency of all transposon mutants in a given gene [30,31]. We considered a gene to have a statistically significant phenotype in a given condition if |gene fitness| > 0.5 and |t| > 3.

Of the 100 carbon sources we tested, 20 compounds met the growth parameters for RB-TnSeq (Fig 1B). 9 of the successful carbon screens involved lipids, which aligns with the well-characterized ability of Mtb to utilize lipids as carbon sources. However, we also conducted screens on 2 sugars (D-glucose and trehalose), glycerol, 4 amino acids (aspartic acid, glutamic acid, asparagine, and casamino acids), and cellular metabolites such as lactate, pyruvate, and malic

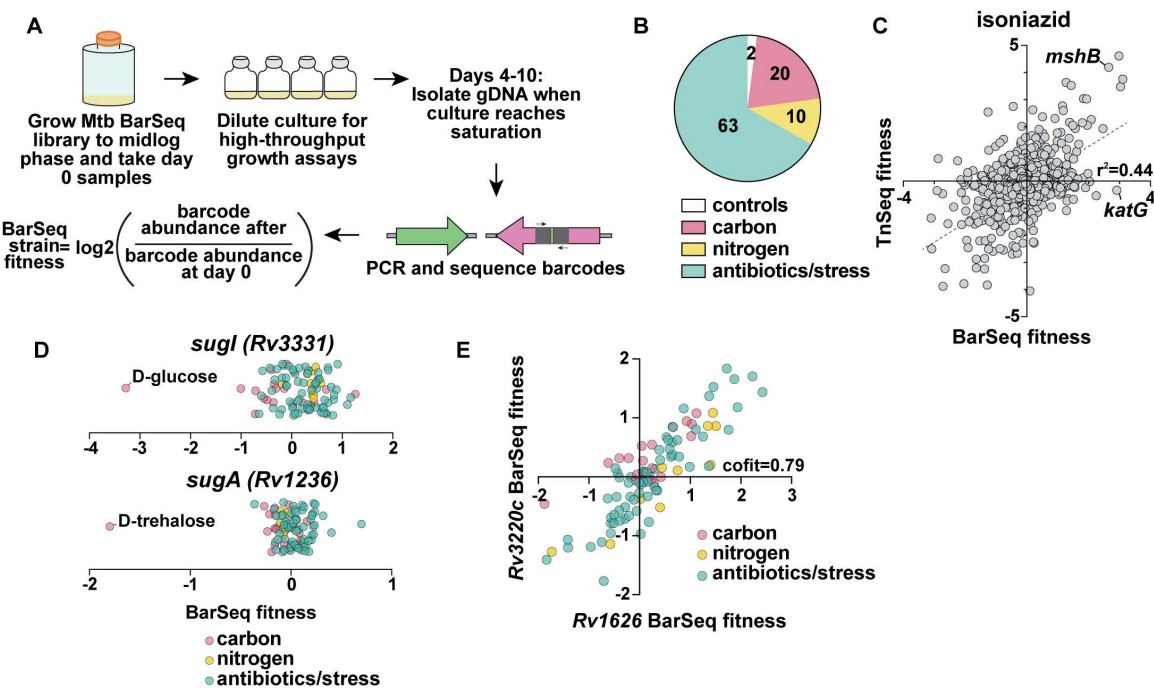

**Fig 1. Overview of RB-TnSeq screens. (A)** Schematic for RB-TnSeq screen experimental set-up. **(B)** Pie chart depicting the number of RB-TnSeq screens completed for each category. **(C)** Previously published TnSeq fitness with 27 ng/mL isoniazid [28] plotted against BarSeq fitness with 25 ng/mL isoniazid. Dotted line represents linear correlation between statistically significant hits from RB-TnSeq with TnSeq. **(D)** BarSeq fitness (log2 fold change) for *sugI* (*Rv3331*) and *sugA (Rv1236)*. **(E)** Cofitness data for *Rv3220c* and *Rv1626*. The data underlying Fig 1B–1E can be found in S1 Data. The RB-TnSeq data for all screens represented in Fig 1B can be found on the public fitness browser (https://fit.genomics.lbl.gov/cgi-bin/org.cgi?orgId=MycoTube) or on FigShare (https://doi.org/10.6084/m9.figshare.30444569).

acid. We did not observe growth on many of the sugars in the chemical library, such as D-fructose, D-xylose, sucrose, D-raffinose, and L-rhamnose. Interestingly, most amino acids did not support growth as a singular carbon source. Of the 43 nitrogen sources we tested, 10 compounds met the growth parameters for RB-TnSeq, comprised of 8 amino acids, urea, and ammonium chloride. Nucleotides, such as uridine, thymidine, inosine, cytidine, thymine, and cytosine, did not result in robust growth as a singular carbon source or nitrogen source. However, these in vitro studies on minimal media do not exclude the possibility that these compounds may support survival of Mtb in vivo. Lastly, of the 131 stressors and antibiotics, 59 were used for RB-TnSeq. Ciprofloxacin, hydrogen peroxide, rifampicin, and clofazimine were tested with two concentrations each, resulting in 63 conditions used for RB-TnSeq that represents 59 unique stressor and antibiotic compounds. Stressors included compounds such as hydrogen peroxide, nitric oxide, metals, and acidic pH. Antibiotics ranged from those not clinically used for TB (e.g., sisomicin, ciprofloxacin, doxycycline), ones traditionally used for TB treatment (e.g., rifampicin, isoniazid, pyrazinamide, and ethambutol), as well as new TB drugs (e.g., pretomanid, dela-manid, and bedaquiline). We also performed controls alongside all screens—either a "7H9 no stress control", our standard media for mycobacterial growth, or "7H9 with 1% DMSO" as a control for compounds dissolved in DMSO.

In total, we conducted 212 successful RB-TnSeq screens, with each screen representing one replicate of a compound at a given concentration. These 212 successful RB-TnSeq screens are comprised of 20 carbon sources, 10 nitrogen sources, and 63 antibiotic and stressor conditions in two biological replicates (Fig 1B). It is important to note that for high-throughput screening, we conducted two replicates for each condition. However, additional replicates can be con-ducted in the future for generating additional high-confidence hits for any condition of interest. The "7H9 no stress control" was performed with each experimental batch and was therefore repeated 24 times. From these data, we calculated 7,524 total statistically significant hits in 850 unique genes. Excluding controls and accounting for compounds conducted with multiple concentrations, only 7 of the 89 unique compounds tested in our RB-TnSeq screens have pre-existing TnSeq data, to our knowledge. These include glycerol [22], cholesterol [22], acidic pH (pH 4.5) [25], and treatment with the anti-biotics isoniazid, rifampicin, ethambutol, and meropenem [28]. When comparing our RB-TnSeq data to these previously published TnSeq results, we observe strong concordance in gene-level phenotypes. Most genes with the strongest phe-notypes are shared across the previously published TnSeq and our RB-TnSeq screens on isoniazid (Fig 1C), rifampicin, meropenem, and ethambutol (S2 Fig) [28]. For instance, mshB (Rv1170), encoding a mycothiol synthesis protein, confers resistance to isoniazid according to both TnSeq and RB-TnSeq, supporting previous findings connecting mycothiol syn-thesis with isoniazid resistance [34]. Mutations in katG (Rv1098c), encoding a catalase-peroxidase known for activating the isoniazid prodrug, was identified as a top hit for isoniazid resistance in the RB-TnSeq screen yet was not a hit in the published TnSeq screen (Fig 1C). Mutants in pirG (Rv3810), encoding an extracellular repetitive protein, are known to be susceptible to rifampicin [35], which was also a shared hit between TnSeq and RB-TnSeq (S2 Fig). In accordance with its annotation as encoding a beta-lactamase, blaC (Rv2068c) was a shared hit between TnSeq and RB-TnSeq for exposure to meropenem (S2 Fig).

Using our high-throughput fitness data, we first examined specific phenotypes, a metric in which a given gene only has a strong fitness defect (|fitness| > 1 and |t| > 5|) in one or a few conditions [31]. Specific phenotypes can be particularly useful in defining the function of a given gene. For example, Rv3331, annotated as encoding a probable sugar-transport integral membrane protein SugI, was only attenuated for growth on D-glucose (Fig 1D), which supports a specific func-tion as a sugar importer. Although SugI has not been shown to import D-glucose experimentally, it has been predicted by homology to transport monosaccharides [36]. In contrast, the sugA (Rv1236) operon, which is known to recycle the disaccharide trehalose [37], was specifically attenuated for growth on D-trehalose (Fig 1D). Our compilation of fitness data yielded 241 specific phenotypes for approximately 170 unique genes (S5 Table).

We also evaluated cofitness, wherein multiple genes share the same fitness score across many conditions, suggest-ing the genes work together in a similar process or pathway [31]. Cofitness is calculated using the Pearson correlation of the fitness values between two genes [31]. We observed high cofitness values for 161 unique gene pairs, in which

PLOS Biology

the cofitness score was greater than 0.8 (S6 Table). For instance, *Rv3220c* and *Rv1626* encode putative members of a bacterial two-component system [38,39]. These two genes have a cofitness score 0.79, supporting existing biochemical data demonstrating a physical interaction (Fig 1E). In addition to their co-fitness, *Rv3220c* and *Rv1626* are both statistically significant hits in 16 different conditions including resistance to antibiotics like norfloxacin, ciprofloxacin, meropenem, linezolid, and isoniazid and hyper-susceptibility to pyrazinamide and para-aminosalicylic acid. These pleiotropic effects suggest this two-component system may have a global effect on the cell, supporting its previous association with lipid metabolism [39].

## Diverse phenotypes of *pe* and *ppe* genes

*Pe* and *ppe* genes are a gene family unique to mycobacteria and are highly represented in Mtb, comprising 10% of coding capacity of the genome [2]. PE/PPE proteins are defined by conserved proline (P) and glutamic acid (E) residues in the N-terminus. PE/PPE proteins can be secreted as heterodimers to the bacterial surface, where they are hypothesized to participate in nutrient acquisition and modulation of the immune response [40]. Given the significant expansion of *pe* and *ppe* genes in Mtb, many questions remain regarding the exact function of this unique gene family, as many members have yet to be characterized. Of the 168 *pe* and *ppe* genes encoded by Mtb, 154 of them were represented in the Mtb RB-TnSeq library. Our screening results demonstrate that *pe* and *ppe* genes display a wide array of fitness phenotypes across the screening conditions (Fig 2A). We calculated 184 statistically significant hits in 37 *pe* and *ppe* genes (S7 Table). Notably, many *pe* and *ppe* genes had zero to very mild fitness defects. This may support findings of PE/PPE proteins modulating the host immune system, which would not be represented in our in vitro screens. For example, *pe-pgrs33 (Rv1818c)*, which has been reported to interact with TLR2 [41], had zero significant phenotypes on the conditions tested (Fig 2A). It is also possible that PE and PPE proteins may act redundantly or are responsive to infection conditions, complicating the ability to observe strong phenotypes.

Hierarchical clustering revealed that some of the strongest phenotypes shown by the *pe* and *ppe* genes were in *pe19*, *ppe50*, and *ppe51* (Fig 2B). The clustering and high cofitness of these genes support previous findings that PPE51 interacts with PE19 [42]. PPE51 has been shown to be important for uptake of glycerol, glucose, maltose, lactose, and trehalose [42–44]. Our RB-TnSeq data additionally demonstrated mild growth defects for *ppe51* on propionate, L-aspartic acid, and L-glutamine as carbon sources. In addition, we found that mutation of *pe19*, *ppe50*, or *ppe51* resulted in resistance to sisomicin, norfloxacin, ciprofloxacin, lomefloxacin, isoniazid, and copper II chloride. Whether PPE50 interacts with PE19 and PPE51 or these phenotypes are due to a polar effect is yet to be experimentally determined.

Some of the strongest cofitness phenotypes we observed were between *pe9*, *ppe54*, and 14 different *pe-pgrs* genes (Fig 2C). *Pe-pgrs* genes proteins are a subset of *pe* genes that contain polymorphic GC-rich sequences (PGRS) [45]. Transposon mutants in this group of *pe-pgrs* genes display resistance to antibiotics, including isoniazid, moxifloxacin, ethioniamide, ofloxacin, para-aminosalicylic acid, streptomycin, levofloxacin, and linezolid. We hypothesize that this gene cluster also mediates import of these antibiotics into the cell or may mediate permeability of the cell envelope. High cofitness values suggests that these PE-PGRS proteins act in the same pathway. Whether they bind each other, or are important for each other's secretion, or some alternative model is yet to be determined.

## PPE3 and ESX-5 are required for growth on diverse nutrient sources

Given previous work linking PE/PPE proteins to nutrient acquisition, we investigated whether specific *pe* and *ppe* genes are required for optimal growth on carbon or nitrogen sources. Notably, *ppe3* (*Rv0280*) mutants exhibited a growth defect across multiple carbon and nitrogen conditions (Fig 2A). *Ppe3* mutants showed reduced fitness when Mtb was grown on glycerol, D-glucose, D-lactate, L-lactate, propionate, and L-asparagine as sole carbon sources as well as on L-serine and L-asparagine as nitrogen sources (Fig 3A). Importantly, *ppe3* mutants were resistant to several antibiotics, including ofloxacin and moxifloxacin, suggesting that loss of *ppe3* does not create a general permeability defect (Fig 3A). *Ppe3* has been

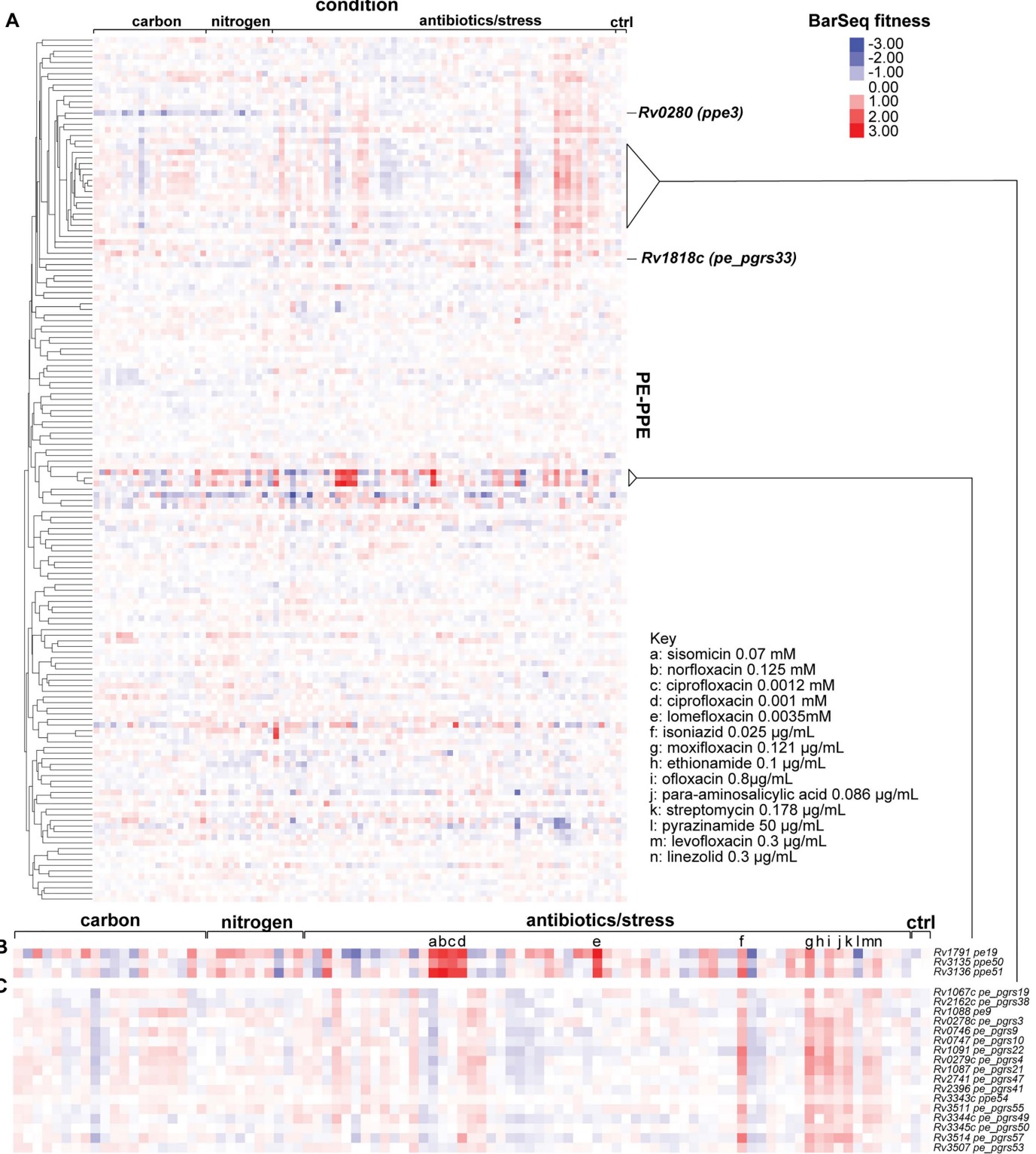

**Fig 2. Heatmap of Mtb *pe and ppe genes*. (A)** Hierarchical clustering of BarSeq fitness values of 154 *pe and ppe* genes represented in the RB-TnSeq library. **(B)** Gene cluster containing *pe19*, *ppe50*, and *ppe51*. **(C)** Gene cluster enriched in *pe-pgrs* genes with high cofitness values. The data underlying this figure can be found in S1 Data.

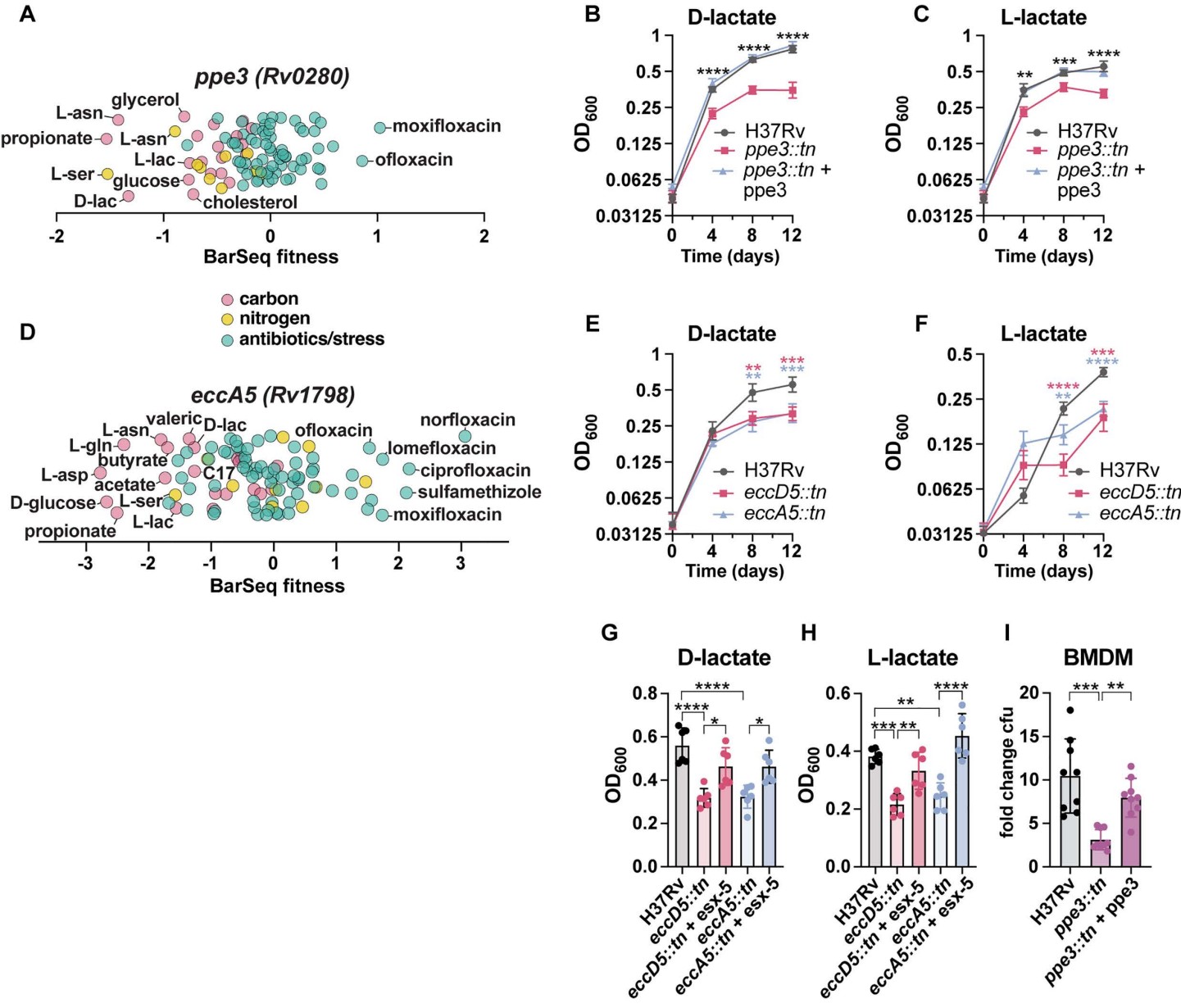

**Fig 3. *ppe3* and *esx-5* are important for growth on nutrient sources.** (A) BarSeq fitness of *ppe3*. (B) OD600 of H37Rv, *ppe3::tn*, and complemented strain on 10mM D-lactate or (C) 10mM L-lactate over 12 days. (D) BarSeq fitness of *eccA5*. (E) OD600 of H37Rv, *eccA5::tn*, or *eccD5::tn* on 10 mM D-lactate or (F) 10 mM L-lactate over 12 days. (G) Complementation of *eccD5::tn* and *eccA5::tn* in 10 mM D-lactate or (H) 10 mM L-lactate on day 12. (I) Fold-change CFU on day 4 from infection of mouse bone marrow-derived macrophages with H37Rv, *ppe3::tn*, and complemented strain. For (B–H), data represent mean±sd for *n* = 6 replicates from two independent experiments. For (F), data represent mean±sd for *n* = 5 replicates from two independent experiments. *p*-values were determined using two-way ANOVA for (B), (C), (E), and (F), and one-way ANOVA for (G) and (H) with Tukey's multiple comparisons test. For (I), data represent mean±sd for *n* = 9 replicates from three independent experiments. *p*-values were determined using Kruskal–Wallis test with Dunn's multiple comparisons test (\**p* < 0.05, \*\**p* < 0.01, \*\*\**p* < 0.001, \*\*\*\**p* < 0.0001). The data underlying this figure can be found in S1 Data.

shown to be upregulated in proteasome mutants and regulated by Zur, a zinc regulator [46,47]. To our knowledge, aside from transcriptional regulation, *ppe3* has not been experimentally characterized. To validate our RB-TnSeq results, we isolated a mutant with a transposon insertion in *ppe3* (*ppe3::tn*) from our arrayed transposon mutant library. This mutant

grew normally in standard 7H9 broth culture (S3 Fig). RB-TnSeq indicated that *ppe3* mutants are unable to grow normally on either L- or D-lactate (Fig 3A). To validate these results, we tested whether the *ppe3::tn* mutant grows when L- or D-lactate is provided as the sole carbon source. We observed that whereas WT Mtb grows robustly on both D and L-lactate, the *ppe3::tn* mutant fails to grow to the same degree as WT on either isomer over the course of 12 days (Fig 3B and 3C). A complementation strain expressing *ppe3* under its native promoter also grew robustly on D- and L-lactate (Fig 3B and 3C). Because the loss of the lipid phthiocerol dimycocerosate (PDIM) results in a growth advantage in Mtb, we verified that the *ppe3* complemented strain and subsequent strains used in this study likely retain PDIM by assaying for sensitivity to vancomycin in the presence of propionate [48] (S4 Fig). Thus, the PPE3 protein is required for utilization of both D- and L-lactate carbon sources.

PPE and PE proteins have been found to be transported to the cell surface by ESX-5, one of the type VII secretion systems encoded by Mtb [49]. Thus, we sought to determine which nutrients require ESX-5 for their utilization. Mutation of the secretory component *eccA5 (Rv1798)* yields negative BarSeq fitness values for propionate, D-glucose, L-lactate, D-lactate, L-serine, acetate, butyrate, L-asparagine, L-glutamine, L-aspartate, valerate, and heptadecanoic acid (Fig 3D). These results are consistent with previous published literature demonstrating that ESX-5 is required for efficient utilization of fatty acids in *Mycobacterium marinum* [50] and for growth on glucose in Mtb [51]. To test whether ESX-5 is indeed required for growth on D- and L-lactate, we isolated Mtb strains with transposon insertions in two different genes of the *esx-5* locus, *eccA5::tn* and *eccD5::tn*. *EccA5* encodes the AAA+ ATPase, and *eccD5* encodes a putative channel protein of the ESX-5 secretion system. These *esx-5* mutants grow at a similar rate as WT in standard growth medium (S3 Fig). Importantly, both mutants fail to grow robustly on both D- and L-lactate over the course of 12 days when compared to WT Mtb (Fig 3E and 3F). Complementation of the *esx-5* operon restored the ability of *eccD5::tn* and *eccA5::tn* to grow on D- and L-lactate (Fig 3G and 3H). *Ppe3* and *eccA5* both exhibited negative BarSeq fitness values during growth on propionate, which is a component of the media in the vancomycin assay to assess PDIM levels. Thus, we further validated that *ppe3::tn*, *eccA5::tn*, *eccD5:tn,* and their complemented strains produce PDIM by performing thin-layer chromatography (TLC) (S5 Fig).

Disruption of ESX-5 in Mtb is known to lead to a virulence defect in macrophages and mice [49]. Given the connection between ESX-5 and PE/PPE proteins and our observation that PPE3 is required for growth on various nutrients, we investigated whether PPE3 is important for growth in host cells. Enumeration of colony-forming units (CFU) from a murine bone marrow-derived macrophage infection revealed that *ppe3::tn* is attenuated for growth in macrophages, which is rescued by the *ppe3* complement. (Fig 3I). Macrophage viability at this day four time point was comparable to uninfected cells and did not vary significantly between the WT, *ppe3::tn*, and the complemented strain (S6 Fig).

## D-lactate is a potential carbon source for Mtb

The shared defect for *ppe3* and *esx5* mutants to grow on D- and L-lactate led us to further investigate lactate utilization in Mtb (Fig 4A). Growth on the distinct D and L isomers of lactate would require conversion into pyruvate by stereo-specific enzymes. The top RB-TnSeq hit for attenuated growth on L-lactate was *lldD2 (Rv1872c)*, which has previously been characterized as an L-lactate dehydrogenase [5] and is under positive evolutionary selection in Mtb clinical strains [6] (Fig 4A). As expected, a transposon mutant in *lldD2* was specifically attenuated for growth on L-lactate but not attenuated for growth on D-lactate or in standard 7H9 growth medium (Figs 4B, 4C, and S3). Expression of *lldD2* in this mutant strain restored growth on L-lactate (Fig 4E).

D-lactate can be produced from lipid and protein metabolism, but is largely produced as a product of carbohydrate metabolism. In macrophages, the glyoxalase system generates D-lactate from methylglyoxal, which is a reactive aldehyde produced as a side-product of glycolysis [52]. However, to our knowledge, D-lactate has yet to be implicated as a carbon source for Mtb. The top RB-TnSeq hit for growth on D-lactate was *Rv1257c*, a putative oxidoreductase (Fig 4A). Using a strain with a transposon insertion in *Rv1257c*, we confirmed that this mutant was specifically attenuated for growth on

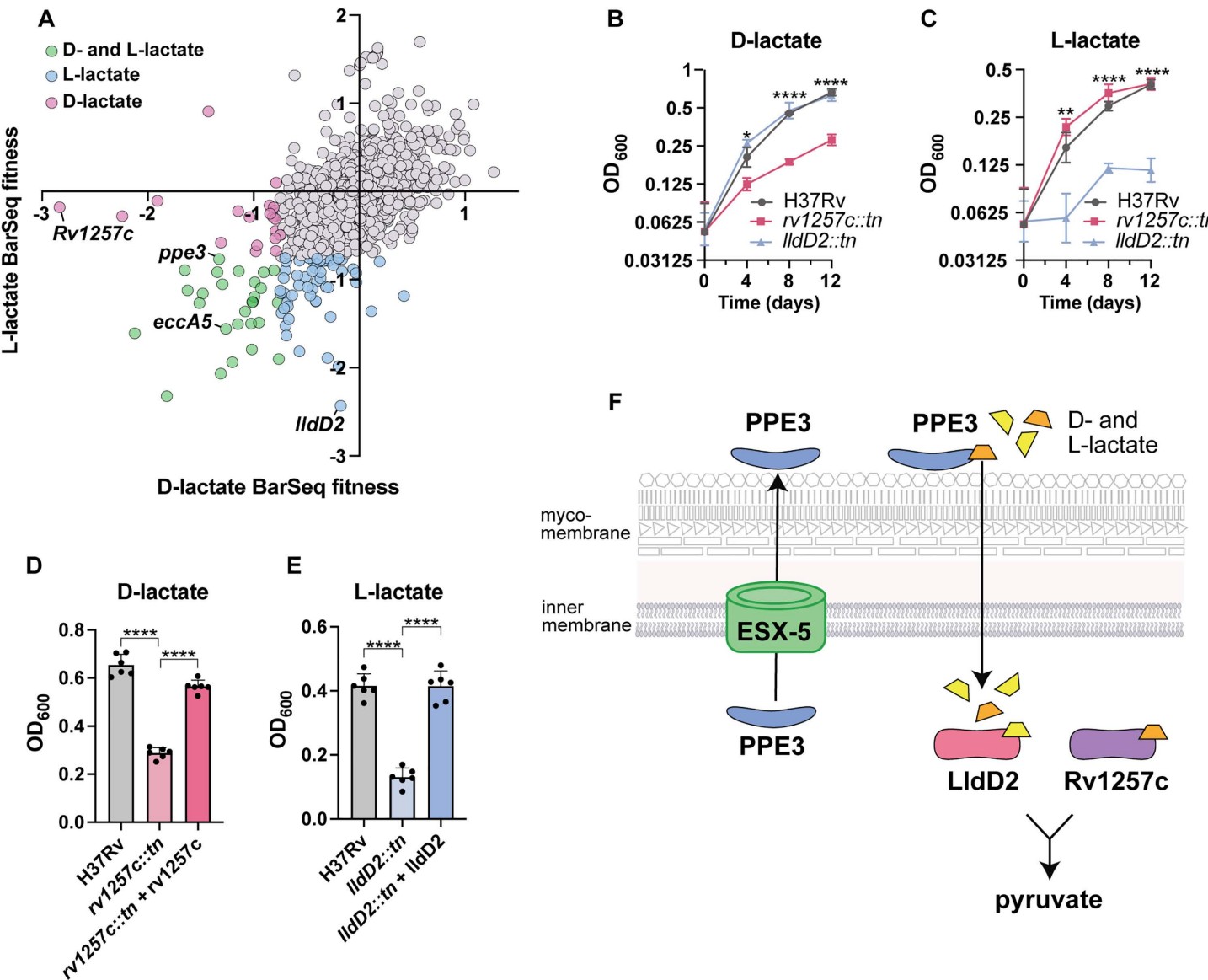

**Fig 4. Stereospecific lactate utilization in lactate dehydrogenase mutants. (A)** L-lactate BarSeq fitness plotted against D-lactate BarSeq fitness. **(B)** $OD_{600}$ of H37Rv, *rv1257c::tn,* and *lldD2::tn* on 10mM D-lactate and **(C)** 10mM L-lactate over 12 days. **(D)** Complementation of *rv1257c::tn* in 10mM D-lactate on day 12. **(E)** Complementation of *lldD2::tn* in 10mM L-lactate on day 12. **(F)** Model for lactate utilization in Mtb. For B-C, data represent mean±sd for *n*=5 replicates from two independent experiments. For D and E, data represent mean±sd for *n*=6 replicates from two independent experiments. *p*-values were determined using two-way ANOVA for (B) and (C) and one-way ANOVA for (D) and (E) with Tukey's multiple comparisons test (*$p<0.05$, **$p<0.01$, ***$p<0.001$, ****$p<0.0001$). The data underlying Fig 4A–4E can be found in S1 Data.

D-lactate (Fig 4B), as it grew to the same $OD_{600}$ as WT on L-lactate and in standard 7H9 growth medium (Figs 4B, 4C, and S3). Complementation restored the ability for the *Rv1257c* mutant to grow on D-lactate (Fig 4D).

Rv1257c contains a potential FAD oxidase domain and shares 40.1% amino acid sequence identity with *E. coli* glycolate oxidase subunit D (*glcD*), which is a member of a protein complex that oxidizes glycolate to glyoxylate. *E. coli* glycolate oxidase uses D-lactate as a substrate at similar kinetics to glycolate [53]. Thus, we predict that Rv1257c acts

as a D-lactate dehydrogenase in Mtb. Thus, RB-TnSeq has allowed us to propose a model where ESX-5 secretes PPE3 or other factors required for the import of D- and L-lactate. After D and L-lactate are imported into the cell, each isomer is converted to pyruvate by stereospecific lactate dehydrogenases (*lldD2* or *Rv1257c*) to enter central carbon metabolism (Fig 4F).

## The *nuo* operon is required for growth on propionate

Lipids are a prominent carbon source for Mtb during infection, so we conducted RB-TnSeq screens on a variety of fatty acid carbon chain lengths, ranging from acetate (C2) to stearate (C18). Metabolism of branched-chain fatty acids, odd-chain fatty acids, and cholesterol results in the production of the three-carbon molecule propionyl-CoA. Propionyl-CoA is oxidized to pyruvate by the methylcitrate cycle through the action of methylcitrate synthase (*Rv1131, prpC),* methylcitrate dehydratase (*Rv1130, prpD*), and methyl-isocitrate lyase (*Rv1129c, icl1*) [54] (Fig 5A). Accordingly, top hits that conferred susceptibility to growth on propionate included *prpC*, *prpD*, and their regulator, *prpR* (Fig 5B).

Another top hit for growth on propionate was the 14-subunit NADH dehydrogenase complex encoded by the *nuo* operon (*nuoA* (*Rv3145)-nuoN* (*Rv3158*)) (Fig 5C), which was not a hit for growth on the two-carbon fatty acid acetate (Fig 5D). Mtb possesses three NADH dehydrogenases; *nuo, ndh* and *ndhA*. Nuo is a classical type I NADH dehydrogenase that oxidizes NADH to NAD+ and contributes to the proton motive force that drives ATP production by translocating protons to the outer membrane space. Ndh (Rv1854c) and NdhA (Rv0392c) are classified as type II NADH dehydrogenases, consisting of one subunit that lacks proton-pumping capabilities.

Using mutants with transposon insertions in *nuoA*, *nuoE*, *nuoG*, or *nuoM*, we confirmed that *nuo* mutants are attenuated for growth on 0.1% propionate as the sole carbon source by measuring $OD_{600}$ over the course of 8 days (Fig 5E and 5F). Complementation of *nuoG::tn* with the *nuo* operon partially restored growth on propionate (Fig 5E). As predicted by the RB-TnSeq data, the *nuo* mutants grew to the same degree as WT on 0.1% acetate after 8 days in culture (Fig 5G). Due to propionate toxicity, bacteria that cannot metabolize propionate, such as methylcitrate cycle mutants, fail to grow in the presence of propionate and acetate [54]. We therefore tested for defects of the *nuo* mutants on a combination of 0.1% acetate and 0.1% propionate. We observed that the growth of *nuo* mutants on this mixed carbon source was attenuated compared to WT and mimicked the growth on 0.1% acetate alone (Fig 5H). Given that the *nuo* mutants can grow on acetate in the presence of propionate, we suspect that propionate is detoxified in the *nuo* mutants but cannot be used as a carbon source to its full potential. Although *nuoG* is a known virulence factor and *nuoG* mutants are attenuated in mouse infection models [55,56], this report is the first to identify a growth defect on propionate, which may contribute to the previously observed virulence attenuation.

## Mutations in *Rv1634* confer resistance to pretomanid

In addition to nutrients, we conducted RB-TnSeq screens to find genes that confer susceptibility and resistance to antibiotics and stressors. Pretomanid is a relatively new TB therapy that is included in new regimens for drug-resistant tuberculosis. It is administered as a prodrug that requires activation by deazaflavin-dependent nitroreductase (Ddn), an enzyme involved in the redox cycling of the cofactor F420 [57]. Mutations that confer resistance to pretomanid have largely been found in the genes that are responsible for activating pretomanid prodrug. These include *ddn*, F420 synthesis genes (*fbiA-D*), and *fgd1*, which encodes an enzyme that reduces the F420 cofactor [58–61]. In alignment with these previous findings, top hits for pretomanid-resistant mutants in the RB-TnSeq screen included *ddn*, the *fbi* operon, and *fgd1* (Fig 6A).

Besides the genes involved in prodrug activation, we find that *Rv1634*, encoding a putative drug efflux transporter, is the strongest hit conferring resistance to pretomanid (Fig 6A). *Rv1634* is operonic with *uvrB (Rv1633)*, an excinuclease in nucleotide excision repair that removes bulky lesions from DNA (Fig 6B). The RB-TnSeq data suggest that *uvrB* and *Rv1634* specifically confer resistance to pretomanid and not any other antibiotics tested in our screens (Fig 6C). Transposon mutants in *uvrB* and *Rv1634* (*uvrB::tn and rv1634::tn*) showed increased survival in the presence of pretomanid as

PLOS Biology

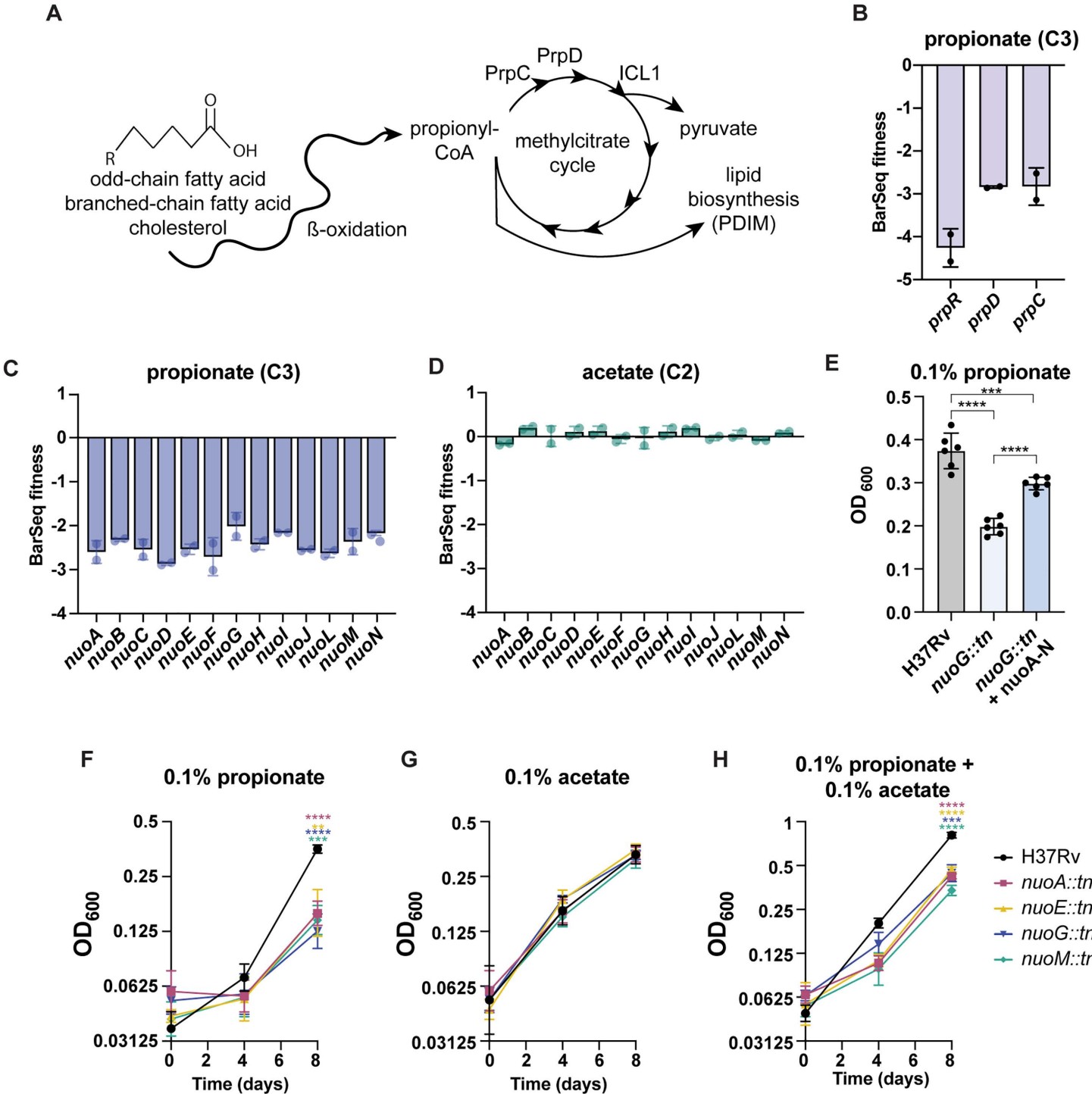

**Fig 5. *Nuo* mutants are attenuated on propionate. (A)** Schematic of propionate detoxification by the methylcitrate cycle or shuttling into lipid biosynthesis. **(B)** BarSeq fitness of methylcitrate cycle genes for growth propionate. **(C)** BarSeq fitness of the *nuo* operon for growth on propionate and **(D)** acetate. **(E)** $OD_{600}$ of H37Rv, *nuoG::tn,* and complemented strain in 0.1% propionate on day 8. Data represent mean±sd for $n=6$ replicates from two independent experiments. **(F)** $OD_{600}$ of H37Rv and *nuo* transposon mutants for growth on 0.1% propionate **(G)** 0.1% acetate, and **(H)** 0.1% propionate and 0.1% acetate on days 0, 4, and 8. Data represent mean±sd for $n=4$ replicates from two independent experiments. *p*-values were determined using one-way ANOVA for (E) and two-way ANOVA for (F), (G), and (H) with Tukey's multiple comparisons test (*$p<0.05$, **$p<0.01$, ***$p<0.001$, ****$p<0.0001$). The data underlying Fig 5B–5H can be found in S1 Data.

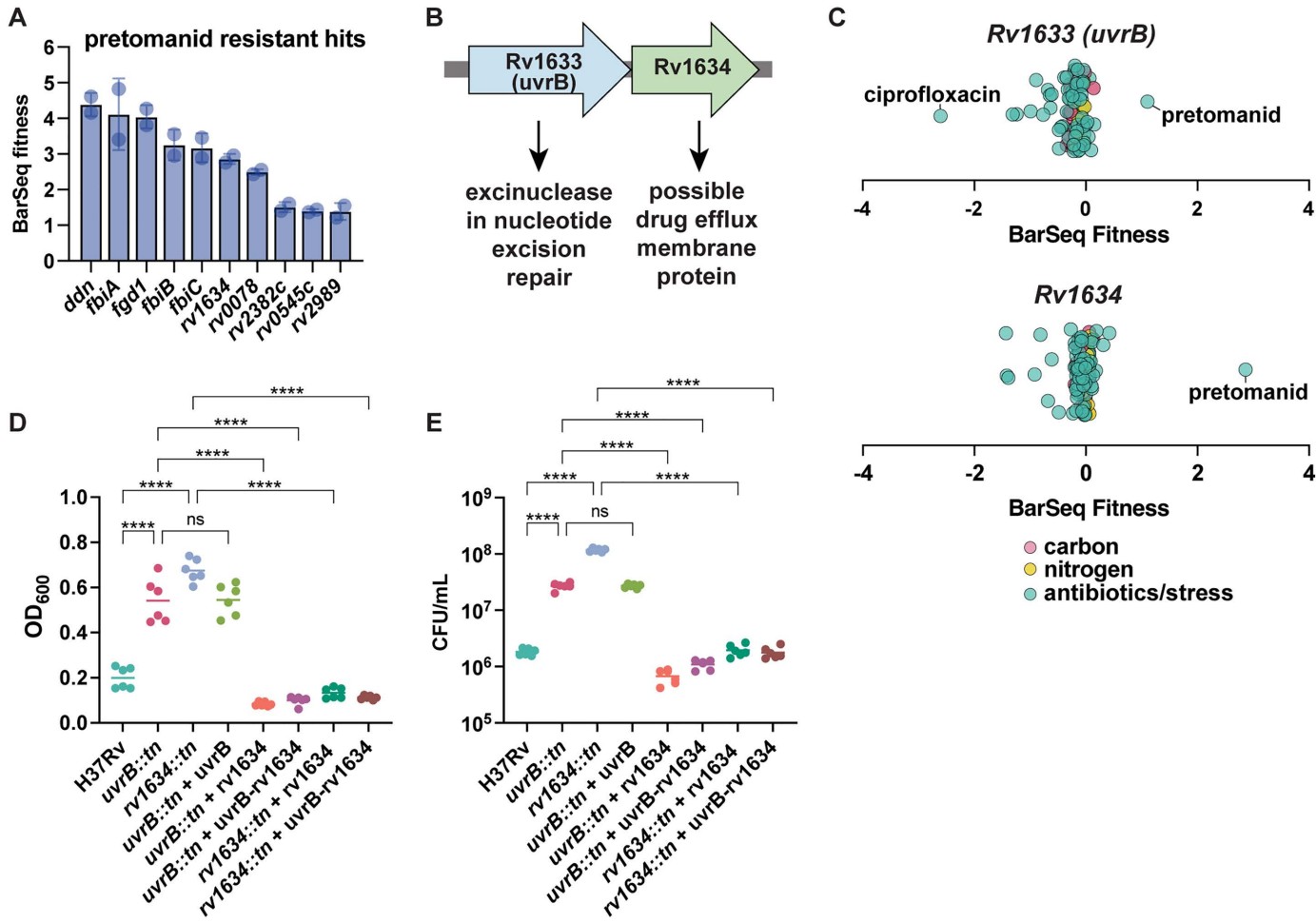

**Fig 6. *Rv1634* confers resistance to pretomanid.** **(A)** Top 10 BarSeq hits conferring pretomanid resistance. **(B)** *uvrB* and *Rv1634* (possible drug efflux membrane protein) are hypothesized to be in an operon. **(C)** BarSeq fitness of all RB-TnSeq screens for *uvrB* and *Rv1634*. **(D)** $OD_{600}$ or **(E)** CFU enumeration of H37Rv, *uvrB::tn*, and *rv1634::tn* and complemented strains after exposure to 0.45 µg/mL pretomanid for 5 days. Data represent mean ± sd for $n = 6$ replicates from two independent experiments. *p*-values were determined using one-way ANOVA with Tukey's multiple comparisons test (**p* < 0.05, ***p* < 0.01, *** *p* < 0.001,*****p* < 0.0001). The data underlying Fig 6A and 6C– 6E can be found in S1 Data.

measured by $OD_{600}$ (Fig 6D) or when plating for CFU after 5 days of antibiotic exposure (Fig 6E), while untreated strains grew similar to WT (S3 and S7 Figs). The CFU data demonstrate that *uvrB::tn* and *rv1634::tn* display approximately a 15-fold and 65-fold increase in pretomanid resistance, respectively. Complementation of *uvrB::tn* with *uvrB* alone failed to restore growth to WT levels in the presence of pretomanid (Fig 6D and 6E). However, complementation of *uvrB::tn* with *Rv1634* or *uvrB-Rv1634* successfully restored susceptibility to pretomanid (Fig 6E). Thus, it appears pretomanid resistance is driven by *Rv1634*, which may confer resistance to pretomanid outside of the conventional *ddn*, *fgd1*, F420 redox recycling pathway typically associated with pretomanid-resistant mutants discovered thus far.

## Discussion

We performed 212 successful RB-TnSeq screens representing 95 conditions on a range of carbon sources, nitrogen sources, stressors, and antibiotics related to Mtb physiology. These high-throughput data allows generation of

hypotheses about gene functions by applying specific phenotypes and cofitness measurements. From our screening data, we dissect a lactate utilization pathway whereby ESX-5 and PPE3 may facilitate D- and L-lactate uptake into the bacterial cell. Lactate is then converted to pyruvate by stereospecific lactate dehydrogenases and enters central carbon metabolism. We also show that the NADH dehydrogenase encoded by the *nuo* operon is essential for growth on propionate. Lastly, we uncover a novel gene conferring resistance to pretomanid, *Rv1634*, a putative drug efflux pump.

A major limitation of our RB-TnSeq screens is that the library is unsaturated due to barcode reuse. The RB-TnSeq library technically contains a large amount of transposon insertions, but many of these barcodes are not usable because they map to multiple insertion sites. Maintaining large barcode diversity throughout the library construction process presented an enormous challenge, and thus the RB-TnSeq library construction was considerably more difficult than that of a standard TnSeq library. Because Mtb exhibits low electroporation and transposition efficiencies, we implemented a phage-based strategy. However, significant loss of barcode diversity occurred during ligation of phAE159 cosmid with pMtb_NN1, as well as during the electroporation of the resulting phagemid into *Mycobacterium smegmatis* for lytic phage production. Given the technical difficulty of the ligation step, we optimized electroporation efficiency and scaled up the number of electroporations performed. This approach enabled construction of an RB-TnSeq library representing ~87% of non-essential genes in Mtb.

Even if the RB-TnSeq library were saturated with usable barcodes, we would lack representation of essential genes, which comprise ~15%–20% of the genome. To address this caveat of transposon libraries, CRISPRi libraries have been developed in Mtb and screened against TB antibiotics [62]. Unlike transposon-based mutagenesis strategies, CRISPRi enables tunable and partial repression of essential genes. Similar to RB-TnSeq, CRISPRi is largely scalable because the measurement of the population is PCR-amplicon-based. However, CRISPRi often requires significant optimization, including the guide design, degree of knockdown, and control of off-target effects [63–65]. Nonetheless, novel insights from these high-throughput genetic platforms will provide a useful resource for deepening our understanding of mycobacterial genetics.

Our observation that Mtb grows robustly on D-lactate in broth as the sole carbon sources reveals a potential for D-lactate to be utilized during infection. D-lactate dehydrogenases have long been elusive in Mtb. To our knowledge, the sole enzyme with D-lactate dehydrogenase activity to have been characterized in Mtb is the flavohemoglobin FHb (Rv0385), which was shown biochemically to oxidize D-lactate to pyruvate by the action of FAD and heme cofactors [66]. A follow-up study on FHb revealed an additional biochemical function as a disulfide oxidoreductase [67]. In our dataset, *Rv0385* did not have significant BarSeq fitness defect for growth on D-lactate. Based on sequence homology to *E. coli*, we predict that *Rv1257c* may be the predominant D-lactate dehydrogenase in Mtb for utilization of D-lactate as a carbon source. For utilization of L-lactate, our findings match previous data that *lldD2* rather than *lldD1* is the primary L-lactate dehydrogenase in Mtb [5]. Interestingly, both our dataset and previously published cholesterol TnSeq screen [22] indicate that *lldD1* may be important for cholesterol utilization.

An interesting question is why Mtb encodes three different NADH dehydrogenases: *nuo*, *ndh*, and *ndhA*. Until recently, *ndh,* a non-proton-pumping NADH dehydrogenase, was thought to be an essential gene because attempts at knocking out *ndh* were unsuccessful [68], and *ndh* was not represented in TnSeq libraries, including our RB-TnSeq library. However, *ndh* knockouts have now been generated in the CDC1551 strain [56] and in H37Rv in fatty-acid-free media [69]. Transposon insertions have been found in the other non-proton-pumping NADH dehydrogenase *ndhA* [21], although questions remain as to the exact roles of these three NADH dehydrogenases. Notably, *ndhA* was not a BarSeq hit for any screen conducted thus far. We hypothesize that Nuo is important when the cell is under certain types of metabolic stress because its proton-pumping capabilities allow for energy conservation due to the contribution to the proton motive force. It is also possible that proton pumping may help modulate the cytoplasmic pH. It is well-appreciated that Mtb is a metabolically flexible bacterium, and these data highlight how the apparent redundancy of three NADH dehydrogenases allows for

growth on different carbon sources. It would be interesting to add radiolabeled propionate to bacterial cultures and track the fate of the radiolabeled carbon in the *nuo* transposon mutants.

The last phenotype we followed up on was concerning the TB antibiotic pretomanid, which is included in new regimens for drug-resistant TB. We found that mutants in *Rv1634* confer resistance to pretomanid, which we hypothesize may act as a stepping-stone to develop even higher resistance in vivo. *Rv1634* encodes a putative drug efflux membrane protein. This presents a counter-intuitive situation. Typically, a loss-of-function mutation in a drug efflux protein drives susceptibility to an antibiotic. However, we observe that a mutation in *Rv1634* drives resistance to pretomanid. We predict that pretomanid may enter the cell through *Rv1634.* Alternatively, the absence of *Rv1634* may induce drug efflux pumps or another drug tolerance mechanism. Future experiments measuring intracellular pretomanid levels or conducting suppressor screens would be enlightening in determining the mechanism of *Rv1634*-driven pretomanid resistance. A deeper understanding of the functions of previously uncharacterized genes in Mtb will be imperative in the development of novel TB antibiotic therapies.

## Materials and methods

### Ethics statement

All procedures involving the use of mice were approved by the University of California, Berkeley Institutional Animal Care and Use Committee (AUP-2015-09-7979-3). All protocols conform to federal regulations, the National Research Council's Guide for the Care and Use of Laboratory Animals, and the Public Health Service's Policy on Humane Care and Use of Laboratory Animals.

### Bacterial strains and culture

The Mtb strain H37Rv was used for generation of the BarSeq library and for all subsequent experiments. Transposon mutants (*ppe3::tn, rv1257c::tn*, *lldD2::tn*, *eccA5::tn, eccD5::tn, nuoA::tn, nuoE::tn, nuoG::tn, nuoM::tn, uvrB::tn, rv1634::tn*) were picked from an arrayed transposon library generated at the Broad Institute. For the ESX-5 complements, pMV-ESX-5 encoding the ESX-5 operon was used as described previously [70]. For the *nuo* complement, the entire *nuo* operon and its upstream promoter was cloned into pMV306. The remaining complemented strains were generated by Gibson assembly cloning of the gene coding region plus 1,000 bp upstream into pMV306. For all experiments, Mtb was grown to midlog phase in Middlebrook 7H9 (Difco BD) liquid medium supplemented with 10% albumin-dextrose-saline, 0.4% glycerol, and 0.05% Tween-80. When plated, Mtb was grown on solid 7H10 agar supplemented with Middlebrook OADC (BD Difco) and 0.4% glycerol. For growth on carbon and nitrogen sources, Sauton's media was prepared as previously described [71], except tyloxapol was substituted for Tween-80. For carbon source experiments in Sauton's minimal medium, glycerol was omitted and replaced by the respective carbon source. For nitrogen source experiments in Sauton's minimal media, asparagine and ammonium iron citrate were omitted and replaced by the respective nitrogen source.

### Generation of the random barcode transposon-site sequencing library

**PacI digestion.** Ten μg of phAE159 cosmid and 10 μg of pMtb_NN1 were PacI digested overnight at 37 °C followed by heat inactivation for 10 min at 75 °C. phAE159 digestion was purified by addition of 1:10 volume of 3M sodium acetate and 2.5 volumes of 100% ethanol. Mixture was incubated at −20 °C for 10 min. DNA was pelleted, washed with 70% ethanol, and air-dried for 5 min. Pellet was resuspended in 15 μL TE buffer. pMtb_NN1 digestion was purified by DNA cleanup and concentrator kit (Zymo Research) and eluted in 15 μL of TE buffer.

**T4 ligation.** 3 μg digested phAE159 cosmid and 1.5 μg digested pMtb_NN1 were incubated with 100 units of T4 Ligase, HC at 16 °C for 4 hours. The reaction was ethanol precipitated as described above and resuspended in 20 μL TE buffer.

**In vitro *lambda packaging and E. coli transduction*.** Ten µL of ligated phAE159 + pMtb_NN1 was added to 25 µL of MaxPlax Packaging extract and incubated at 30 °C for 90 min. Another 25 µL of MaxPlax Packaging extract was added to reaction mixture and incubated at 30 °C for another 90 min. 500 µL of PD buffer (10 mM Tris-HCl pH 8.3, 100 mM NaCl, and 10 mM $MgCl_2$) was added to stop the reaction followed by addition of 25 µL chloroform. The packaging reaction was then titered to estimate the efficiency of the reaction. Packaged lambda phage was incubated with Stbl3 *E. coli* at 37 °C for 1.25 hours with no shaking and gentle vortexing every 15 min. Bacteria were then pelleted at 3,500 rpm for 5 min and resuspended in LB broth. Bacteria were plated on 24.5 cm² LB + 50 µg/mL kanamycin plates to achieve 60,000 CFU per plate. A total of $1 \times 10^6$ CFUs were plated. The next day, colonies were collected by scraping into LB broth and phagemid was purified by midi-prep.

**Electroporation of BarSeq phagemid into M. smegmatis to generate mycobacteriophage.** Electrocompetent *M. smegmatis* was prepared by washing 2 L of *M. smegmatis* grown to $OD_{600} = 0.2$ with ice cold 10% glycerol and resuspending in a final volume of 20 mL 10% ice cold glycerol. 400 µL of electrocompetent *M. smegmatis* and 200 ng of phagemid were added to 0.2 cm electroporation cuvettes. Cuvettes were incubated on ice for 10 min before electroporation pulse of 2.5 kV, 25 µF, and 1,000 Ω. 2 mL of LB broth was immediately added to the electroporation cuvette, transferred to 15 mL conical tubes, followed by incubation at 37 °C for 2 hours with no shaking. 8 mL of top agar (2 mM $CaCl_2$, 0.6% agarose, melted and cooled to 55 °C) was added, and seeded onto two 7H10 plates. Plates were incubated at 30 °C for 3 days for plaque formation. A total of 100 electroporations were conducted to make the RB-TnSeq library. We found that preparing *M. smegmatis* for electroporation at a low $OD_{600}$ (around ~0.1–0.2 rather than ~0.5–1) and adding a recovery step before plating in top agar improved electroporation efficiency.

**Mycobacteriophage collection and concentration.** Phage was collected by adding 3 mL MP buffer (50 mM Tris pH 7.6, 150 mM NaCl, 10 mM $MgCl_2$, and 2 mM $CaCl_2$) to each 7H10 plate and rocked at 4 °C overnight. The liquid on each plate was collected, pooled into 50 mL conical tubes, and centrifuged at 4,000 rpm for 20 min to pellet any bacteria. The supernatant was then passed through a 0.22 µm filter. To concentrate phage, a 20% PEG-8000/2.5M NaCl mixture was incubated with the phage at a 7.5 mL PEG-NaCl to 30 mL phage ratio on ice for 2 hours. The phage was pelleted at 15,000 rpm for 30 min at 4 °C. Phage was resuspended in MP buffer and subsequently titered.

**Mycobacterium tuberculosis transduction.** 1 L of Mtb was grown to $OD_{600} = 0.8$, washed two times with MP buffer, and resuspended in a final volume of 9 mL MP buffer. 1 mL of phage (concentration between $1 \times 10^{11}$ and $1 \times 10^{12}$ pfu/mL) was added to the bacteria and incubated at 37 °C for 18 hours without shaking. Transduction was then washed two times and resuspended in 10 mL PBS + 0.05% Tween80. The transduction was plated on 7H10 + 50 µg/mL kanamycin + 0.05% Tween80 24 cm² plates and incubated at 37 °C for 21 days. Colonies were collected by scraping into 7H9 broth. Bacterial clumps were disbursed by sonication and light vortexing, and the RB-TnSeq library was aliquoted into cryovials for storage at −80 °C.

## RB-TnSeq fitness assays

Mtb RB-TnSeq library was grown from frozen stocks in 7H9 + 50 µg/mL kanamycin in roller bottles for 5 days at 37 °C. For day 0 barcode abundance, 5 mL of culture was pelleted and stored at −80 °C until genomic DNA extraction. For carbon source experiments, the culture was washed 2× with Sauton's minimal media minus carbon (no glycerol). For nitrogen source experiments, the culture was washed 2× with Sauton's minimal medium minus nitrogen (no asparagine and ammonium iron citrate). For antibiotic and stress experiments, the culture was washed 2× with 7H9. Cultures were started at $OD_{600} = 0.05$ in 10 mL in inkwells in duplicate and shaken at 37 °C. For carbon and nitrogen sources, the cultures were pelleted for genomic DNA extraction when the culture reached saturation ($OD_{600} = 0.8$–1) or had at least doubled three times ($OD_{600} > 0.4$). For antibiotic and stress experiments, cultures were pelleted on day 5. For antibiotic and stress experiments that were more than 50% inhibitory (e.g., acidic pH, see S4 Table), cultures were outgrown in regular 7H9 and were pelleted when the culture reached midlog phase.

## Mtb genomic DNA extraction

10 mL of culture was pelleted and frozen at −80 °C. When thawed, pellet was resuspended in 440 μL RB buffer (25 mM Tris-HCl pH 7.9, 10 mM EDTA, and 50 mM D-glucose) with 1 mg/mL lysozyme and 0.2 mg/mL RNase A and incubated at 37 °C overnight. 100 μL 10% SDS and 50 μL 10 mg/mL proteinase K were added and incubated at 55 °C for 30 min. 200 μL 5M NaCl was added followed by gentle mixing. 160 μL of pre-heated cetrimide saline solution (4.1 g NaCl and 10 g of cetrimide (hexadecyltrimethylammonium bromide) in 90 mL $H_2O$) was added and incubated at 65 °C for 10 min. 1 mL of chloroform:isoamyl alcohol (24:1) was added and inverted to mix. Mixture was centrifuged at 14,000 rpm for 10 min, and 800 μL of aqueous layer was transferred to a new tube. 560 μL (0.7× volume) isopropanol was added and mixed to precipitate DNA. Tubes were centrifuged at 14,000 rpm for 10 min, DNA pellet was washed with 700 μL 70% ethanol, and centrifuged at 14,000 rpm for 5 min. Pellets were air-dried and covered in 50 μL water. DNA was stored at 4 °C overnight to allow for DNA pellets to dissolve.

## Growth curves for carbon sources

Mtb strains were grown to midlog phase in 7H9, washed two times with Sauton's medium minus carbon, and diluted to $OD_{600} = 0.05$ in 10 mL in inkwells in media containing the respective carbon source (10 mM sodium D-lactate, 10 mM sodium L-lactate, 0.1% sodium propionate, 0.1% sodium acetate). Cultures were shaken at 37 °C, and $OD_{600}$ was measured on days indicated in figure legends. All carbon sources are from Sigma-Aldrich.

## Determination of antibiotic IC50s in 96-well plates

Mtb strains were grown to midlog phase in 7H9 and washed two times with fresh 7H9. Mtb was added to 96 well-plates containing 2-fold dilutions of antibiotics for a final starting $OD_{600}$ of 0.05 in 100 μL. Plates were incubated in a standing incubator at 37 °C in a sealed Tupperware with a water-wet rag to prevent evaporation for 10 days. $OD_{600}$ was measured using a SpectraMax M2 Microplate Reader. IC50s were determined by a nonlinear regression on GraphPad Prism.

## Determination of pretomanid resistance in culture

Mtb strains were grown to midlog phase in 7H9 and washed two times with fresh 7H9. Mtb was added to inkwell bottles containing 0.45 μg/mL pretomanid (Sigma-Aldrich) for a final starting $OD_{600}$ of 0.05 in 10 mL. Cultures were shaken at 37 °C for 5 days. CFUs were determined by diluting bacteria into PBS + 0.05% Tween-80 and plating serial dilutions on 7H10.

## Bone marrow-derived macrophage infections

Bone-marrow macrophages were collected from mice through a protocol approved by the UC Berkeley Institutional Animal Care and Use Committee (IACUC), protocol AUP-2015-09-7979-3. Femurs from C57BL/6 mice were flushed and resulting cells were cultured in Dulbecco's Modified Eagle Medium (DMEM) supplemented with 10% fetal bovine serum (FBS) and 10% supernatant from 3T3-M-CSF cells for 6 days. Two days before infection, BMDMs were seeded at a concentration of 50,000 cells per well in a 96-well plate. Bacteria were prepared by washing two times with PBS, sonicated, and spun at 500 rpm for 5 min to pellet clumps. Bacteria were diluted into DMEM supplemented with 5% FBS and 5% horse serum at a multiplicity of infection of 1. Following a 4-hour phagocytosis period, infection medium was removed, and cells were washed with room temperature PBS before fresh medium was added. For CFU enumeration, medium was removed and cells were lysed in water with 0.5% Triton-X and incubated at 37 °C for 10 min. Following the incubation, lysed cells were resuspended and serially diluted in PBS with 0.05% Tween-80. Dilutions were plated on 7H10 plates.

## Cell viability assay

The 96-well plate with bone marrow-derived macrophages was equilibrated to room temperature for 30 min. Following a PBS wash, 100 μL of Cell Titer Glo 2.0 (ProMega) was added per well. The plate was incubated for 10 min at room temperature. Luminescence was measured in a white opaque 96-well plate with a SpectraMax M2 Microplate Reader.

## Van-10-P assays

Van-10-P assays were conducted as described previously [48]. Briefly, Mtb strains were inoculated into 96-well plates at a starting $OD_{600}$ = 0.005 in 7H9 with 0.1 mM sodium propionate and 0.05% tyloxapol with or without 10 µg/mL vancomycin (ThermoScientific). Plates were incubated for 10 days at 37 °C at which the $OD_{600}$ was measured using a SpectraMax M2 Microplate Reader. Van-10-P growth % was calculated as the ratio of the $OD_{600}$ of vancomycin-treated wells over the untreated wells. Mtb Erdman lacking PDIM (*fadD28::tn*) was used as a negative control and described previously [72].

## PDIM lipid extraction and TLC

Mtb cultures were grown to an $OD_{600}$ of 0.5–1 in standard 7H9 with 0.05% tyloxapol substituted for 0.05% Tween-80. A total of 25 OD for each strain was pelleted by centrifugation at 3,500 rpm for 5 min, resuspended in 1 mL hexanes, and incubated at room temperature for 5 min. Cultures were centrifuged at 3,500 rpm for 5 min and the hexane supernatant was removed and added to a tube with 1 mL chloroform and 2 mL methanol. An additional 1 mL of chloroform and 1 mL of $H_2O$ were added, vortexed, and centrifuged at 3,500 rpm for 5 min. The bottom (organic) layer was transferred to a new tube. The solvent was evaporated by nitrogen gas and lipids were resuspended in 200 µL 2:1 chloroform:methanol. 10 µL of lipid extract was spotted on an aluminum-backed silica TLC plate (SiliCycle) and run in 98:2 petroleum ether:acetone. 5 µL (8 µg) of purified PDIM from H37Rv was spotted as a positive control and obtained through BEI Resources, NIAID, NIH:NR-20328. The TLC plate was subsequently dipped in $H_2O$, stained for 15 min in 0.03% Coomassie blue (Bio-Rad) in 25% methanol, followed by destaining for 15 min in 25% methanol.

## Data analysis

TnSeq sequencing and data analysis, BarSeq, computation of gene fitness values, t scores, specific phenotypes, cofitness, and quality metrics were done as described previously [30]. Genes were considered to have significant BarSeq fitness if the |log2 fold change| > 0.5 and |*t*-like statistic| > 3. We slightly relaxed the *t*-like statistic threshold in this study (|*t*| > 3 rather than |*t*| > 4) due to the considerably smaller size of the Mtb RB-TnSeq library compared to other RB-TnSeq libraries [31]. The standard analysis pipeline for RB-TnSeq does not normalize fitness values by the number of generations for each condition. In this study, the number of doublings varied modestly across conditions (~3–4.5 generations). It has been previously demonstrated that normalization of fitness values by generation time does not meaningfully impact the ability to identify significant phenotypes [31]. Hierarchical clustering of *pe* and *ppe* genes was computed in Gene Cluster 3.0 and visualized in Java TreeView. All other statistics were calculated in GraphPad Prism 10. For *p*-values, \*$p < 0.05$, \*\*$p < 0.01$, \*\*\*$p < 0.001$, \*\*\*\*$p < 0.0001$.

## Supporting information

**S1 Table. Essential genes from TnSeq data.**
(XLS)

**S2 Table. Carbon sources tested in this study.**
(XLSX)

**S3 Table. Nitrogen sources tested in this study.**
(XLSX)

**S4 Table. Antibiotics and stressors tested in this study.**
(XLSX)

**S5 Table. Specific phenotypes.**
(XLSX)

**S6 Table. Genes with high cofitness.**
(XLSX)

**S7 Table. *Pe* and *ppe* genes with statistically significant BarSeq hits.**
(XLSX)

**S1 Fig. Protocol for construction of RB-TnSeq library in Mtb.** Temperature-sensitive phAE159 cosmid was combined with pMtb_NN1 containing the transposase, *Himar1* mariner barcoded transposons, and a kanamycin resistant cassette via PacI digest and concatemer ligation. The ligated cosmid was packaged into lambda phage and transduced into *E. coli* to be midi-prepped. Barcoded phagemid was electroporated into *Mycobacterium smegmatis* and incubated at 30 °C for lytic phage plaque formation. Once the phage was collected and concentrated, it was transduced into Mtb and plates were incubated at 37 °C. After 21 days, colonies were scraped from plates and pooled to create the RB-TnSeq library.
(TIF)

**S2 Fig. TnSeq vs. RB-TnSeq.** Previously published TnSeq fitness [28] plotted against RB-TnSeq fitness for rifampicin, meropenem and ethambutol. Dotted line and $r^2$ represent linear correlation between hits with log 2 fold change $> 0.5$ from RB-TnSeq with TnSeq. The data underlying this figure can be found in S1 Data.
(TIF)

**S3 Fig. Growth curves of transposon mutants in 7H9.** (A–E) Growth of transposon mutants and complemented strains in standard growth media (7H9) measured by $OD_{600}$ over the course of 5 days. (F) Doubling time calculated from growth curves of transposon mutants and complemented strains. Data represent mean $\pm$ sd for $n = 6$ replicates from two independent experiments. The data underlying this figure can be found in S1 Data.
(TIF)

**S4 Fig. Van-10-P assay with transposon mutants and complements.** Van-10-P assay as a proxy for the PDIM levels for strains used in this study. WT Erdman and an Erdman strain lacking PDIM (*fadD28::tn*) are positive and negative controls, respectively. Some transposon mutant strains are slightly more susceptible to vancomycin compared to WT but are highly enriched compared to the PDIM-minus strain (*fadD28::tn).* Data represent mean $\pm$ sd for at least $n = 6$ replicates from two independent experiments. For **(A)**, *p*-values were determined using Welch's *t* test. For **(B–F)**, *p*-values were determined using one-way ANOVA with Dunnett's multiple comparisons test (*$p < 0.05$, **$p < 0.01$, ***$p < 0.001$,****$p < 0.0001$). The data underlying this figure can be found in S1 Data.
(TIF)

**S5 Fig. PDIM TLC for *ppe3* and *esx-5* strains.** TLC lipid analysis of PDIM levels in H37Rv, *ppe3::tn*, *eccD5::tn, eccA5::tn* and complemented strains. Purified PDIM and *fadD28::tn* are used as positive and negative controls, respectively. TLC is representative of two biological replicates.
(TIF)

**S6 Fig. Cell viability during Mtb infection of mouse bone marrow-derived macrophages.** Bone marrow-derived macrophage viability measured by cell titer glo on day 4 after infection with H37Rv, *ppe3::tn*, and ppe3 complemented strain. Data represent mean $\pm$ sd for $n = 9$ replicates from three independent experiments. *p*-values were determined using one-way ANOVA with Dunnett's multiple comparisons test and were all nonsignificant. The data underlying this figure can be found in S1 Data.
(TIF)

**S7 Fig. Untreated controls of pretomanid-resistant strains and complements. (A)** $OD_{600}$ and **(B)** CFU of untreated H37Rv, *uvrB::tn,* or *rv1634::tn* and complemented strains on day 5. Data represent mean $\pm$ sd for $n = 6$ replicates from

two independent experiments. $p$-values were determined using one-way ANOVA with Tukey's multiple comparisons test. (\*$p<0.05$, \*\*$p<0.01$, \*\*\*$p<0.001$,\*\*\*\*$p<0.0001$). The data underlying this figure can be found in S1 Data. (TIF)

**S1 Data. Numerical data and statistical analysis for all figures.**
(XLSX)

**S1 Raw Images. Raw image of TLC from S5 Fig.**
(PDF)

## Acknowledgments

We thank Morgan Price for help with data analysis. We thank Edith Houben for the ESX-5 complementation plasmid. We thank members of the Stanley, Cox, and Vance labs for helpful discussions about this work.

## Author contributions

**Conceptualization:** Kayla M. Dinshaw, Katie A. Lien, Sorel V. Yimga Ouonkap, David F. Savage, Adam M. Deutschbauer.

**Data curation:** Kayla M. Dinshaw, Sarah A. Stanley.

**Formal analysis:** Kayla M. Dinshaw.

**Funding acquisition:** David F. Savage, Sarah A. Stanley.

**Investigation:** Kayla M. Dinshaw, Katie A. Lien, Matthew Knight, Sorel V. Yimga Ouonkap, Hualan Liu, Hans K. Carlson, Adam M. Deutschbauer.

**Methodology:** Kayla M. Dinshaw, Sorel V. Yimga Ouonkap, Hualan Liu, Hans K. Carlson, Adam M. Deutschbauer.

**Project administration:** Adam M. Deutschbauer, Sarah A. Stanley.

**Resources:** Sarah A. Stanley.

**Supervision:** Adam M. Deutschbauer, Sarah A. Stanley.

**Writing – original draft:** Kayla M. Dinshaw, Sarah A. Stanley.

**Writing – review & editing:** Kayla M. Dinshaw, David F. Savage, Hans K. Carlson, Adam M. Deutschbauer, Sarah A. Stanley.

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
