## [Editor Report · Decision Letter 0]

12 Nov 2025

Dear Dr Stanley,

Thank you for submitting your manuscript entitled "Random barcode transposon-site sequencing in Mycobacterium tuberculosis reveals the functions of uncharacterized genes" for consideration as a Methods and Resources by PLOS Biology.

Your manuscript has now been evaluated by the PLOS Biology editorial staff, as well as by an academic editor with relevant expertise, and I am writing to let you know that we would like to send your submission out for external peer review.

Once your full submission is complete, your paper will undergo a series of checks in preparation for peer review. After your manuscript has passed the checks it will be sent out for review. To provide the metadata for your submission, please Login to Editorial Manager (https://www.editorialmanager.com/pbiology) within two working days, i.e. by Nov 14 2025 11:59PM.

Kind regards,

Melissa

Melissa Vazquez Hernandez, Ph.D.

Associate Editor

PLOS Biology

---

## [Decision Letter · Decision Letter 1]

12 Jan 2026

Dear Dr Stanley,

Happy New Year!

Thank you for your patience while your manuscript "Random barcode transposon-site sequencing in Mycobacterium tuberculosis reveals the functions of uncharacterized genes" went through peer-review at PLOS Biology. Your manuscript has now been evaluated by the PLOS Biology editors, an Academic Editor with relevant expertise, and by four independent reviewers.

As you will see in the reports, all reviewers are really positive about the study, as we are. However, a few concerns and suggestions were raised that we would encourage you to address. Reviewer 1 calls the study the most comprehensive M. tuberculosis TnSeq-style dataset to date, but flags the need to benchmark RB-TnSeq results against existing large TnSeq collections and to quantify how incomplete library saturation affects essentiality/fitness calls. Reviewer 2 requests to temper/adjust the title and claims (RB-TnSeq facilitates, but doesn’t alone “reveal” functions), to clarify statistical thresholds and nomenclature, and to address a potential fitness-calculation limitation. Reviewer 3 requests some validation analysis such as confirming PDIM status to avoid confounding PPE/ESX-5-linked nutrient phenotypes and avoiding unsupported statements about ESX-5 secretion of PPE3. Reviewer 4 asks for more transparency on the technical hurdles and trade-offs of RB-TnSeq versus conventional TnSeq and CRISPRi, so others can adopt the approach.

In light of the reviews, which you will find at the end of this email, we are pleased to offer you the opportunity to address the comments from the reviewers in a revision that we anticipate should not take you very long. We will then assess your revised manuscript and your response to the reviewers' comments with our Academic Editor aiming to avoid further rounds of peer-review, although we might need to consult with the reviewers, depending on the nature of the revisions.

**IMPORTANT - SUBMITTING YOUR REVISION**

*Resubmission Checklist*

*Published Peer Review*

*PLOS Data Policy*

*Blot and Gel Data Policy*

Sincerely,

Melissa

Melissa Vazquez Hernandez, Ph.D.

Associate Editor

PLOS Biology

REVIEWERS' COMMENTS

Reviewer #1:

This manuscript describes the generation and application of a random barcode transposon-site sequencing (RB‑TnSeq) library in Mycobacterium tuberculosis to systematically define gene function across diverse growth conditions. The authors performed 95 RB‑TnSeq screens encompassing a wide range of carbon and nitrogen sources, stress conditions, and antibiotics. Using single-strain experiments and genetic complementation, they validate phenotypes for several mutants (e.g., ppe3, eccD5, eccA5, lldD2, rv1257c, nuoG). Key findings include: (i) M. tuberculosis can use D‑lactate as a carbon source and this utilization requires rv1257c; (ii) growth on D/L‑lactate depends on ppe3 and the ESX‑5 secretion system; (iii) the type I NADH dehydrogenase encoded by the nuo operon is required for optimal growth on propionate; and (iv) inactivation of rv1634 confers resistance to pretomanid.

This work presents the most comprehensive TnSeq dataset for M. tuberculosis to date, and the study is both well executed and clearly presented. There are, naturally, additional lines of investigation the authors could pursue; for example, biochemical demonstration that rv1257c functions as a D‑lactate dehydrogenase would further strengthen the conclusions. In this reviewer's opinion, however, such experiments fall beyond the reasonable scope of the current manuscript.

The main area for improvement is in the comparison with existing datasets. The MtbTnDB database, for instance, contains more than 150 TnSeq screens, several of which were carried out under conditions similar to those tested here. It would be helpful to show how the RB‑TnSeq dataset compares with and complements these prior studies, and to what extent the new fitness and essentiality calls are consistent with previous determinations.

The authors rightly note that incomplete saturation of the RB‑TnSeq library is a "major limitation," but the manuscript does not clearly convey the magnitude of this issue. Given that M. tuberculosis gene essentiality has already been mapped by conventional TnSeq with high resolution and confidence, a direct comparison between RB‑TnSeq‑derived essentiality calls and publicly available datasets could provide an empirical estimate of false positive and false negative rates associated with the RB‑TnSeq library.

Reviewer #2:

The manuscript by Dinshaw and colleagues describes the use of random barcode transposon-site sequencing (RB-TnSeq) to facilitate the functional assessment of characterized and as yet uncharacterized genes in Mycobacterium tuberculosis. As described, RB-TnSeq enabled the fitness analysis of 2,888 genes in 212 screens across 95 unique experimental conditions. This challenging approach enabled the identification of hundreds of novel phenotypes and correlation of genotype/phenotype within and across conditions. Further validation studies confirmed novel roles for genes involved in nutrient utilization, stress tolerance and drug susceptibility/resistance. Importantly, the authors present their data in a versatile easily searchable web-based resource that will be of great value to the microbiology community. Overall, the manuscript has a clear and logical flow and the work is rigorous with well-supported conclusions.

As described below, there are some points, mostly editorial in nature, that should be considered to improve the manuscript.

Major:

1. The title "Random barcode transposon-site sequencing in Mycobacterium tuberculosis reveals the functions of uncharacterized genes" overrepresents what is provided by RB-TnSeq alone where validation was required to fully reveal functions of uncharacterized genes. It also does not capture many of the other novel genotype/phenotype associations that were discovered. I would recommend modifying the title to something involving RB-TnSeq as facilitating high-throughput elucidation of gene function across diverse conditions.

2. BarSeq fitness: The approach as described seems limited in its ability to provide highly reproducible fitness values between studies. The major limitation is due to not including an expansion factor, a critical experimental variable, as part of the fitness calculation. Without including population expansion as part of the calculation, "fitness" values will vary between experiments not necessarily because of actual fitness differences, but because there was a difference in the amount of growth of the population (and respective mutants). For example, say the generation time of the population is 24 hours, and a mutant of interest has a generation time of 30 hours. Using the described approach, if the population is grown for 72 hours it will expand 8-fold (log2 fold abundance change from t0 of 0), the mutant will expand 5.76 fold (log2 fold abundance change from t0 of -0.47). If the population is grown for 120 hours it will expand by 32-fold (log2 fold abundance change from t0 of 0), the mutant will expand by 16-fold (log2 fold abundance change from t0 of -1). By not accounting for expansion, the apparent fitness of the mutant changes with the amount of growth of the population, however, in this example the mutant's fitness is the same and it is the expansion that is a variable that is unaccounted for. It is not clear whether this limitation is addressable in the current study, and if not it should be described as a limitation in the discussion in the paragraph that starts on line 461.

3. Lines 166-168, While the statistical analysis was based on previous studies, the authors need more information and clarification on their approach and why they chose specified thresholds, which seem to differ from the cited reports which both use |t|>4 versus |t|>3 of the current study. Also, please indicate how this corresponds to threshold for FDR.

4. There are numerous statements throughout the manuscript that confuse gene versus protein designations which detract from the flow and overall quality of the writing. Some of these errors are noted below, however, there are many more that the authors should identify and correct.

Line 117, change " a mycobacterial gene family..." to "encoding a mycobacterial protein family..."

Line 207, change "a mycothiol" to "encoding a mycothiol"

Line 209, Change "Mutants in katG (Rv1098c), a catalase-peroxidase" to "Mutations in katG (Rv1098c), encoding a catalase-peroxidase"

Line 212, Change as above.

Line 214, change "a beta-lactamase" to "encoding a beta-lactamase"

Lines 216-218, the respective statement belongs in the discussion.

Lines 223-224, change "as a" to "as encoding a" and change sugI to SugI.

Line 225, change "sugI" to "SugI" and correct all subsequent nomenclature errors where genes and proteins are not appropriately distinguished.

5. Y-axis of plots in Figures 3B, C, E, F, 4B, C, 5F, G, H should be logarithmic since it will facilitate comparison of exponential growth rates. Authors should also comment on differences in growth rates and yields since carbon utilization is a major aspect of the study.

Minor:

Lines 77-80, Stating patient non-adherence as the driving factor for the rise in MDR- and XDR-TB is incomplete and can be viewed as offensive to some. I recommend modifying "resulting in patient non-adherence" to "contributing to treatment disruption".

Lines 91-92, It is true that there are many unannotated genes (as yet unidentified open reading frames) in Mtb, however, those genes with the annotation of "hypothetical protein" are indeed annotated. Unannotated genes were not assessed in the described approach since the approach was dependent upon gene annotation. As such, it would be better to say that these "genes remain uncharacterized with many being annotated as encoding hypothetical proteins and others with functional annotations based on homology with orthologs from other species."

Lines 92-94, It is not clear how points in this sentence fit with the story and they were not address, I recommend deleting.

Line 98, "infer" is an over-representation of what is learned by TnSeq. Change "infer" to "better understand", or something of the like.

Line 124, change "for drug resistant Mtb strains." to "for treatment of drug resistant Mtb infections."

Lines 141-143, These statements on difficulty, loss of diversity at each step and optimization refer to data that is not shown. Please either include as supplemental data or omit this statement.

Lines 163-164, Modify "Mutants in genes..." to "Bar codes associated with a growth advantage..." or something of the like.

Line 171, change "were" to "involved"

Line 184, change "conducted" to "administered", or something of the like

Line 312, Figure S4 does not show testing of all strains in the VAN-P-10 assay. Please modify statement accordingly. Also, since PDIMs were not directly measured, it would be best to say "likely retain PDIM".

Line 415, Change "Mutants" to "Mutations"

Line 441-447, should move this to discussion.

Line 452, "This high-throughput data allows" to "These high-throughput allow"

Line 508, change "mutant" to "mutation"

Line 510-511, An alternative possibility is that in the absence of Rv1634 function, a different efflux or other drug tolerance mechanism could be induced.

Line 510, change "mutant" to "mutation"

Line 535, Was a different iron salt used? If not, phenotypes could be associated with a change in iron abundance.

Line 581, change "conicals" to "conical tubes"

Line 595, change "dissolved" to "disbursed"

Reviewer #3:

The manuscript "Random barcode transposon-site sequencing in Mycobacterium tuberculosis reveals the functions of uncharacterized genes" (PBIOLOGY-D-25-03612R1) describes the development of a barcoded mutant library in M. tuberculosis (Mtb) and how the library was used in 20 different "BarSeq fitness" selection screens.

Barcoded mutant libraries have been developed in numerous other bacterial systems to make unbiased fitness screens cheaper and easier to complete. This work set out to construct a barcoded library in Mtb. Due to technical difficulties, the barcoded library generated in these studies contained 2,888 individual transposon mutants with each carrying unique barcodes. Unfortunately, this is not a completely saturating mutant library. However, the library does cover ~85% of the non-essential genes. This strain of Mtb has ~4,000 genes and ~625 of these genes are considered essential or are associated with a negative fitness defect when mutated and the bacteria are plated on standard media (PMID: 28096490).

Next, the authors surveyed various nutrients and stresses to identify conditions that yielded sufficient bacterial growth or death to facilitate positive/negative selection studies with the barcoded libraries. The team settled on 89 or 95 conditions (see below) and conducted duplicate "BarSeq fitness" screens by comparing the input mutant barcodes to the mutant barcodes obtained after the selective pressure. In the end, the data revealed 850 unique genes that were enriched or depleted across all of the conditions tested.

As a QC check the manuscript highlights genes identified in this screen that have previously been associated with INH resistance (katG) and B-lactam resistance (blaC). Additional examples that the approach is accurate are found in the supplement S4 table. For example, genes associated with: ethionamide resistance (VriS/Rv3083), clofazimine resistance (Rv0678), and cholesterol import/metabolism (Mce4) were accurately called with the method giving confidence for the approach. The data in this work also supports previously known data indicating that PE19, PPE50, PPE51 are involved in nutrient uptake in Mtb.

Consistent with previous reports in the literature that PE/PPE proteins can be involved in nutrient import in Mtb, the BarSeq data also revealed that ppe3 mutants have reduced fitness on various nutrients (glycerol, glucose, lactate, propionate, asparagine). This was confirmed using an isolated mutant and demonstrating that this strain has a growth defect in media containing D- and L-lactate as a sole carbon source.

Numerous PPE/PE proteins are transported to the Mtb cell surface by the ESX-5 system and a secretory protein in this pathway eccA5 displayed a growth defect on numerous nutrient sources in the BarSeq data. Based on this the team isolated an ESX-5 mutant (eccA5:tn) and confirmed that this mutant has a fitness defect for growth on D- and L-lactate. Further, this manuscript provides convincing evidence that Mtb can metabolize and grow on lactate in a LldD2 and Rv1257 dependent manner.

The next BarSeq observation that the authors highlight is that the NADH dehydrogenase (nuo) mutants fail to grow in the presence of propionate. This has not been reported before and was confirmed with several isolated nuo mutants (nuoA, nuoE, nuoG, nuoM). It is unclear how the nuo participates in Mtb's handling of propionate.

Lastly, the BarSeq screening identified known and novel mutations in uvrB and rv1634 which correspond to pretomanid (Pa) resistance. The phenotype of isolated uvrB::tn and rv1634::tn mutant confirm the Pa resistance phenotype but it is not clear how these genes are involved.

Overall, this manuscript reads easy and is quite clear. In this reviewer's opinion this paper is not fully a resource paper and not fully a research paper. It is not a full resource because there are many genes missing from the analysis as it is not a saturating library. I found myself looking for my favorite genes in the supplement lists and they are just not there because the mutants are not in the library. While several genes/conditions identified in the screen were followed-up using isolated mutants this follow up was mostly to confirm phenotypes predicted from the screen and little new biology emerged from these studies. I leave it to the editor to make the publication decision.

Major suggestions

Given the reported issues with loss of cell surface PDIM and how this confounds measurements of PPE associated functions in Mtb (Reference #41) and infection studies, I suggest you use a second method to confirm that your WT, ppe3::tn, eccA5::tn, and complemented strains do actually produce PDIM. I suggest this because the ppe3 and eccA5 mutants have a pronounced growth defect in propionate media (Fig 3 and S6 table) suggesting an issue in propionate uptake/metabolism. This is especially important since the Van-10-P assay used to predict the presence of PDIM in your strains (S4 Figure) is dependent on normal levels of propionate import/metabolism by the bacteria.

It Is not demonstrated in this manuscript that ESX-5 secretes PPE3 (lines 454-455). The lactate metabolism defect in the ESX-5 mutants could be explained by the loss of another PPE protein from the bacteria's surface.

Can you provide evidence that PPE3 is secreted in an ESX-5 dependent manner or is this published somewhere?

Minor suggestions

I think line 157 should read "131 stressors and antibiotics" to match data in Table S3. Similar on line 183?

The numbers on line 193 do not seem to match the supplement data, it looks like 89 unique conditions were screened (20 carbon, 10 nitrogen, 59 stress). Please double check this. Maybe explain where the 212 number came from, did some conditions have more replicates?

Line 200, should this read 89 instead of 95?

Line 450, make this consistent with the condition #'s above if it is not accurate.

Line 484, lldD1 has previously been flagged as being required for cholesterol utilization via tnSeq (PMID: 22365605).

Reviewer #4:

The authors develop random barcode transposon sequencing (RB-TnSeq) for Mycobacterium tuberculosis. Although this technique has been used by other laboratories to study a variety of bacterial organisms, it has not previously been applied to M. tuberculosis. A major advantage of RB-TnSeq is that library preparation - often a major bottleneck in traditional TnSeq - can be simplified substantially, relying either on PCR amplification of specific barcodes or deep sequencing of all barcode-associated DNA. This simplification enables TnSeq to be performed at scale. Consistent with this strength, the authors carry out a large number of RB-TnSeq experiments, growing M. tuberculosis on diverse carbon and nitrogen sources and in the presence of multiple anti-TB compounds. Overall, I found the manuscript compelling and generally well executed, the online data to be of use to the field, and I have only a few suggestions.

My primary concern is that, while the authors emphasize the benefits of RB-TnSeq in the Introduction, they also state that library generation presented considerable difficulty. However, these difficulties are not described in sufficient detail in the main text, which may limit the ability of other labs to adopt this approach. The authors note that "significant optimization" was required, but do not elaborate on what aspects of the workflow were challenging or how these challenges were overcome.

Relatedly, the Discussion focuses almost exclusively on the biological findings, and does not address the advantages and disadvantages of RB-TnSeq relative to other functional genomics approaches. For example, how does library generation compare to conventional TnSeq in M. tuberculosis? Why do the authors think it was particularly difficult to generate or maintain library diversity? How does RB-TnSeq compare to CRISPRi-based screens in terms of scalability, coverage, or experimental limitations? While I appreciate the biological insights presented, this omission feels like a missed opportunity. As this study represents the first proof-of-principle application of RB-TnSeq in M. tuberculosis, a more explicit discussion of the technical challenges, trade-offs, and future optimization would help set the stage for broader adoption of this promising approach.

Minor

Lines 159: Is saturation more than 3 doubling as stated in the paragraph above?

Figure 2 - the text is much too small to be legible

---

## [Decision Letter · Decision Letter 2]

9 Mar 2026

Dear Sarah,

Thank you for your patience while we considered your revised manuscript "Random barcode transposon-site sequencing in Mycobacterium tuberculosis facilitates high-throughput characterization of gene function across diverse conditions" for publication as a Methods and Resources at PLOS Biology. This revised version of your manuscript has been evaluated by the PLOS Biology editors, the Academic Editor and the original reviewers.

Based on the reviews, we are likely to accept this manuscript for publication, provided you satisfactorily address the remaining points raised by the reviewer 2. Please also make sure to address the following data and other policy-related requests.

1) We routinely suggest changes to titles to ensure maximum accessibility for a broad, non-specialist readership, and to ensure they reflect the contents of the paper. In this case, we would suggest a minor edit to the title, as follows. Please ensure you change both the manuscript file and the online submission system, as they need to match for final acceptance:

"High-throughput characterization of Mycobacterium tuberculosis gene function across diverse conditions"

2) Please add the weblink of the funding agencies in the Financial Disclosure statement in the manuscript details.

3) Please cite the location of the data clearly in all relevant main and supplementary Figure legends, e.g. “The data underlying this Figure can be found in S1 Data” or “The data underlying this Figure can be found in https://doi.org/10.5281/zenodo.XXXXX”

4) Please ensure that your Data Statement in the submission system accurately describes where your data can be found and is in final format, as it will be published as written there

5) Per journal policy, if you have generated any custom code during the course of this investigation, please make it available without restrictions. Please ensure that the code is sufficiently well documented and reusable, and that your Data Statement in the Editorial Manager submission system accurately describes where your code can be found. More information on our Code Policy, what and how to share can be found here: https://journals.plos.org/plosbiology/s/code-availability

We expect to receive your revised manuscript within two weeks.

*Published Peer Review History*

*Press*

Sincerely,

Melissa

Melissa Vazquez Hernandez, Ph.D.

Associate Editor

PLOS Biology

REVIEWERS' COMMENTS

Reviewer #1:

I fully support acceptance of the revised manuscript.

Reviewer #2:

The authors have done a nice job of addressing comments raised in review. My only additional suggestions is to change "genetic approaches" to "mutation-based genetic approaches" on line 28 of the abstract. The current statement ignores the use of CRISPRi as a large-scale genetic approach to understand gene function.

Reviewer #3:

The comments I made in the previous submission have been addressed. The authors double checked the sections that previously did not seem consistent in this version.

I have no more major suggestions to make.

Reviewer #4:

This manuscript has been significantly improved. I have no further suggestions.

---

## [Editor Report · Decision Letter 3]

24 Mar 2026

Dear Dr Stanley,

Thank you for the submission of your revised Methods and Resources "High-throughput characterization of Mycobacterium tuberculosis gene function across diverse conditions" for publication in PLOS Biology. On behalf of my colleagues and the Academic Editor, Matthew Waldor, I'm pleased to say that we can in principle accept your manuscript for publication, provided you address any remaining formatting and reporting issues. These will be detailed in an email you should receive within 2-3 business days from our colleagues in the journal operations team; no action is required from you until then. Please note that we will not be able to formally accept your manuscript and schedule it for publication until you have completed any requested changes.

Sincerely,

Roli Roberts

Roland G Roberts, PhD

Senior Editor

PLOS Biology

rroberts@plos.org

on behalf of

Melissa Vazquez Hernandez, Ph.D., Ph.D.

Associate Editor

PLOS Biology
